# Long-term statins administration exacerbates diabetic nephropathy via ectopic fat deposition in diabetic mice

Tong-sheng Huang [1,2,4], Teng Wu[1,2,4], Yan-di Wu[1,2], Xing-hui Li[1,2], Jing Tan[1,2], Cong-hui Shen[1,2], Shi-jie Xiong[1,2], Zi-qi Feng[1,2], Sai-fei Gao[1], Hui Li[1] & Wei-bin Cai [1,2,3]

Statins play an important role in the treatment of diabetic nephropathy. Increasing attention has been given to the relationship between statins and insulin resistance, but many randomized controlled trials confirm that the therapeutic effects of statins on diabetic nephropathy are more beneficial than harmful. However, further confirmation of whether the beneficial effects of chronic statin administration on diabetic nephropathy outweigh the detrimental effects is urgently needed. Here, we find that long-term statin administration may increase insulin resistance, interfere with lipid metabolism, leads to inflammation and fibrosis, and ultimately fuel diabetic nephropathy progression in diabetic mice. Mechanistically, activation of insulin-regulated phosphatidylinositol 3-kinase/protein kinase B/mammalian target of rapamycin signaling pathway leads to increased fatty acid synthesis. Furthermore, statins administration increases lipid uptake and inhibits fatty acid oxidation, leading to lipid deposition. Here we show that long-term statins administration exacerbates diabetic nephropathy via ectopic fat deposition in diabetic mice.

According to the International Diabetes Federation, it was estimated that in 2021 there were 537 million people with diabetes worldwide. These figures are expected to increase to 783 million by 2045[1]. Diabetes has become a serious global public health problem. In China and elsewhere, diabetic nephropathy (DN) is a chronic progressive disorder that can lead to end-stage renal disease and result in kidney replacement therapy. DN has become more common than chronic kidney disease (CKD) related to glomerulonephritis in both the general population and a hospitalized urban population in China[2].

It has long been known that individuals with type 2 diabetes mellitus (T2DM) are prone to concurrent hyperlipidemia and almost invariably develop a serious breakdown in lipid dynamics, with elevated levels of circulating free fatty acids (FFAs), cholesterol (CHOL) and triglycerides (TG), along with excessive deposition of fat in various tissues, such as liver, kidney, and muscle[3]. Lipid deposition in non-adipose tissues is known as ectopic fat deposition (EFD)[4]. Hyperlipidemia can result in a progressive decline in kidney function accompanied by albuminuria and a decrease in the glomerular filtration rate (GFR) through the activation of multiple intracellular and biochemical pathways that collectively contribute to the development of glomerulosclerosis, endothelial dysfunction, oxidative stress, mesangial expansion, podocyte loss and albuminuria, and tubulointerstitial damage[5]. Therefore, it is important to control blood lipids in patients with T2DM. Lipid management in patients with T2DM needs to be more stringent than that in nondiabetic patients. According to Kidney Disease: Improving Global Outcomes Clinical Practice Guideline for Lipid Management in CKD, statins treatment has been recommended for lipid management in CKD[6]. In 2019, the guidelines

[1]Guangdong Engineering & Technology Research Center for Disease-Model Animals, Laboratory Animal Center, Zhongshan School of Medicine, Sun Yat-sen University, Guangzhou 510080 Guangdong, China. [2]Department of Biochemistry, Zhongshan School of Medicine, Sun Yat-sen University, Guangzhou 510080 Guangdong, China. [3]Guangdong Provincial Key Laboratory of Digestive Cancer Research, The Seventh Affiliated Hospital of Sun Yat-sen University, Shenzhen 518107 Guangdong, China. [4]These authors contributed equally: Tong-sheng Huang, Teng Wu. ✉e-mail: caiwb@mail.sysu.edu.cn

for the management of dyslipidemia issued by the European Society of Cardiology and European Atherosclerosis Society noted that patients with T2DM who were at very-high risk were recommended to take statins, and the evidence-based level was 1A[7]. According to statistics, 63% of those with diagnosed diabetes reported taking prescription CHOL-lowering medications, which were mainly statins therapy, and the number continues to rise[8].

For a long time, many randomized controlled trials (RCTs) have shown that lowering low-density lipoprotein (LDL) CHOL with statins could safely benefit a wide range of high-risk patients and reduce coronary mortality and morbidity in high-risk patients[9,10]. However, the effects of statins on DN progression remain controversial. Especially in recent years, a large number of studies have reported that statins can exacerbate insulin resistance and interfere with glucose metabolism, and statins users have an increased risk of developing new-onset type 2 diabetes[11]. Although several trials have evaluated the effects of statins on kidney disease outcomes and whether statins could lower the mortality and risk of cardiovascular disease[12], for patients with DN, long-term statins administration may interfere with glucose metabolism and increase insulin resistance, thereby further affecting the progression and prognosis of DN. In recent years, some large-scale RCTs have confirmed that statins therapy did not slow kidney disease progression within 5 years in a wide range of patients with CKD[13]. More importantly, statins with high CHOL-lowering efficacy might increase the risk for developing severe renal failure[14]. Among adolescents with type 1 diabetes, statins therapy over a period of 2–4 years could increase hyperglycemia and diabetic ketoacidosis among adolescents[15]. Therefore, the effects of long-term statins administration on the progression of DN remain controversial. It may be too soon to determine whether long-term statins administration ameliorates the adverse effects of insulin resistance and hyperglycemia exposure.

Based on a large number of clinical trials and basic research, most researchers believe that statins have a protective effect on DN. It should be emphasized that all of the trials and basic research were short in duration and that no data on long-term statins administration for diabetes are available from trials lasting longer than 10 years. Until recently, a retrospective matched-cohort study with 12 years of data on patients with diabetes found that statin use was associated with diabetes progression, including a greater likelihood of insulin treatment initiation, significant hyperglycemia, acute glycemic complications, and an increased number of prescriptions for glucose-lowering medication classes[16]. This finding reminds us that whether the beneficial effects of long-term statins administration on DN outweigh the detrimental effects is unclear.

In this work, we used three kinds of diabetic mice (db/db mice, KK-Ay mice and STZ-induced diabetic mice), used two kinds of statins (atorvastatin and rosuvastatin) and two doses of drugs (5 mg kg⁻¹ and 10 mg kg⁻¹ atorvastatin) to observe the effects of long-term statins administration on DN. Unexpectedly, we found that long-term statins administration accelerated DN progression. Mechanistically, long-term statins administration not only activated the insulin-regulated Phosphatidylinositol 3-kinase (PI3K)/protein kinase B (Akt)/mammalian target of rapamycin (mTOR) signaling pathway, leading to increased renal fatty acid synthesis, but also increase lipid uptake in the kidney and inhibited fatty acid oxidation (FAO), leading to EFD in the kidney.

## Results

### Basis for the dose and duration of statins administration in mice
Our study mainly used db/db mice, and STZ-induced diabetic mice and KK-Ay diabetic mice were used in replicate experiments to validate the results, as described in the schematic diagram (Fig. 1a). The administration time was determined by measuring the UACR and urinary KIM-1 levels (Fig. 2m, n). Based on the results, the endpoint of the experiment was a 50-week duration of diabetes. We not only regularly tested

urinary hemoglobinuria (Supplementary Fig. 1), but also serum CK-MB and myoglobin in the mice at the end of the experiment (Supplementary Table 1), and ruled out that the mice had statin-associated muscle pain (myalgia) and needed to be stopped early. As previously suggested, 5 and 10 mg kg⁻¹ atorvastatin and 20 mg kg⁻¹ rosuvastatin have been repeatedly demonstrated to be effective in lowering lipids in mice[17–19]. According to the literature, our dose produces plasma concentrations that are far lower than those achieved after oral administration of common doses of atorvastatin in humans[20,21]. In addition, we used the classic high-fat diet (HFD) mouse model and apolipoprotein E (ApoE)⁻/⁻ HFD mouse model to determine that 5 and 10 mg kg⁻¹ atorvastatin and 20 mg kg⁻¹ rosuvastatin could significantly reduce blood CHOL, while the lower dose of 3 mg kg⁻¹ had no significant CHOL-lowering effects (Supplementary Figs. 2 and 3). Overall, we determined that the doses of the two statins were safe and effective.

However, there might be a survivor bias, given that the mice were administered statins for 40 weeks, and at this timepoint were <25% of the original statins treated cohort survived. To rule out survivor bias, we added short-term administration experiments, ~10 weeks of statins administration, and measured GFR and kidney damage in the mice. As shown in Supplementary Fig. 4, short-term statins treatment reduced the survival rate of mice (a) and resulted in certain degree of renal injury (b–g). Therefore, we believe that survivor bias may not be the main determinant factor for us to analyze and consider the aggravated renal injury caused by long-term statins treatment in db/db mice.

### Long-term statins administration worsens insulin resistance and increases mortality in db/db mice
We found that db/db mice gradually lost weight after statins administration, and with time, db/db mice in the statins administration groups began to show significant increases in mortality, whereas no increase in mortality was observed in db/m mice that were given statins at the same time (Fig. 1b, c). Subsequently, we found that the fasting blood glucose (FBG) levels of db/db mice in the statin groups were significantly elevated at the end of administration (Fig. 1d), and further testing revealed that GHbA1c was also markedly elevated in the Db + ATO10 and Db + Rosu20 groups (Fig. 1e). The effect on insulin resistance was evaluated by ITT assays, and both statins administration groups showed significant exacerbation of insulin resistance compared to the Db group (Fig. 1f). RAGE expression was assessed by immunohistochemical staining, and the results showed that long-term statins administration notably increased renal AGEs levels (Fig. 1g, h). Collectively, these data suggest that long-term statins administration leads to insulin resistance, exacerbates blood glucose control, and increases the mortality of diabetic mice.

### Long-term statins administration worsens renal injury in db/db mice
Representative images of db/db mouse kidneys were examined, and the kidneys in the statin groups showed atrophy and deformation (Fig. 2a). Representative images of periodic acid schiff (PAS) staining showed typical pathological changes in glomeruli in the statin groups, such as glomerular hypertrophy, visible glomerular mesangial expansion and glomerular basement membrane (GBM) thickening (Fig. 2b, g, h). Moreover, we observed the same pathological changes in KK-ay diabetic mice and STZ-induced diabetic mice (Supplementary Figs. 5b and 6b). The transmission electron microscopy (TEM) results further showed fusion of glomerular podocyte pedicles and glomerular basement thickening (Fig. 2c). We observed periodic acid-silver methenamine (PASM) staining and found that the glomerular capillary basement membrane in the statins administration groups was significantly thickened, part of the basement membrane was broken, capillary loops were expanded, and the mesangial matrix was expanded (Fig. 2d). In addition, the statins groups showed increased diabetes-induced podocyte loss (Fig. 2e, i); at the molecular level, the

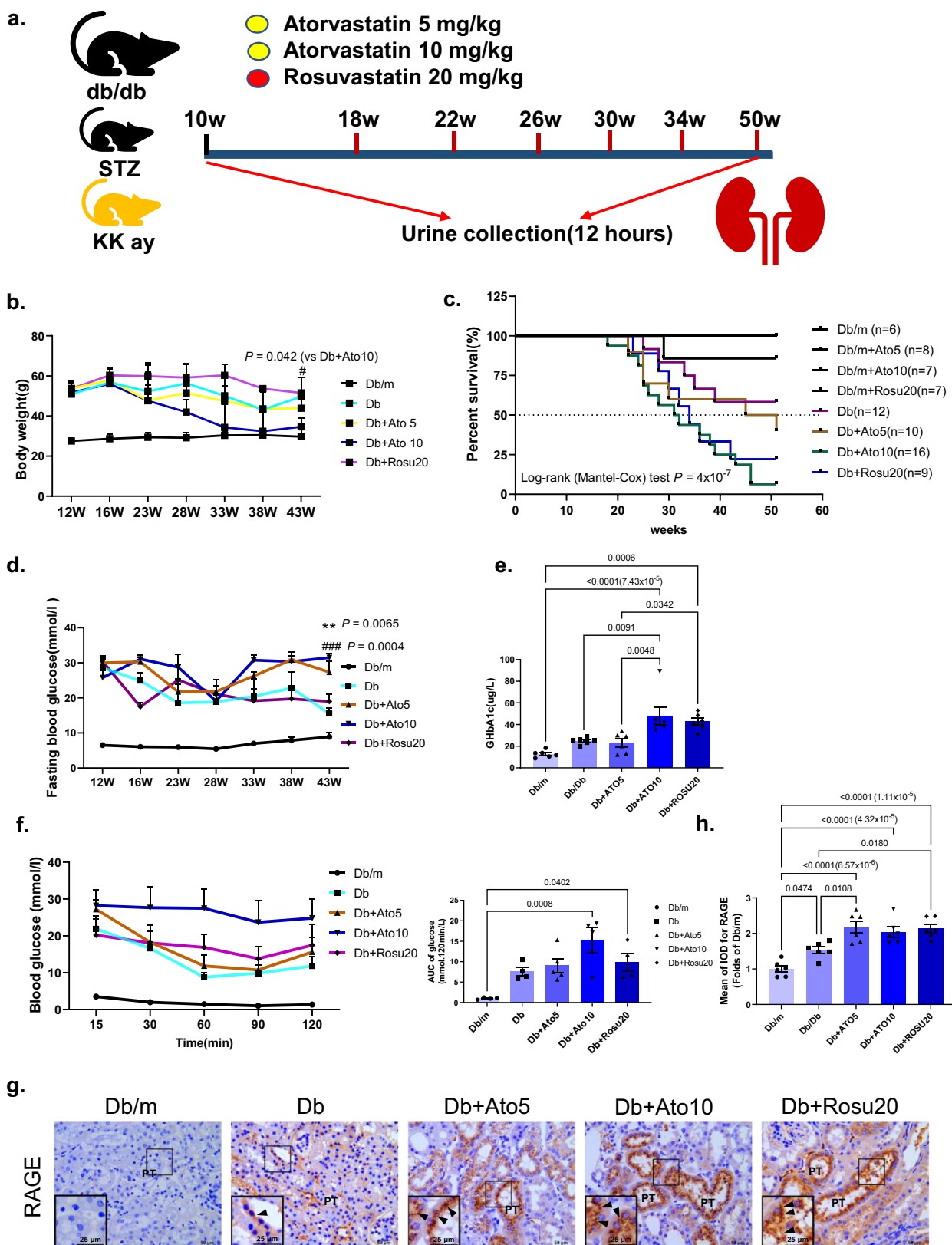

podocyte-associated protein, Nephrin was significantly down-regulated, compared to that in the Db group (Fig. 2f, j). PAS and Masson staining showed no significant pathological changes in *db/m* mice after long-term statins administration (Supplementary Fig. 7a, b). Therefore, we no longer examine the kidney of *db/m* mice after long-term statins administration, and *db/m* mice without statins treatment were used as the control group.

The UACR and urinary KIM-1 levels are classic indicators that are used to assess diabetic kidney injury, and the UACR is commonly used to assess early diabetic kidney injury and glomerular damage.

**Fig. 1 | Long-term statins administration worsens insulin resistance in *db/db* mice.** All mice were ~50 weeks old. **a** Schematic diagram showing the procedure of long-term administration of statins for *db/db* mice. **b** Body weight. For 12 weeks, *n* = 6 biologically independent mice in Db/m group, *n* = 13 in Db group, *n* = 10 in Db + Ato5 group, *n* = 16 in Db + Ato10 group, *n* = 9 in Db + Rosu20 group. But over time, some of the mice gradually lost weight and died. By 43W, *n* = 6 biologically independent mice in Db/m group, *n* = 6 in Db group, *n* = 5 in Db + Ato5 group, *n* = 3 in Db + Ato10 group, *n* = 3 in Db + Rosu20 group. For specific values see as a "Source Data file". **c** Kaplan–Meier survival curves of *db/db* mice after long-term administration of statin. Log-rank (Mantel–Cox) test was used for the analysis of statistical significance. Exact *p* value = $4 \times 10^{-7}$. **d** Measurements of fasting blood glucose. *n* = 4 per time point. Significance **$p < 0.01$ compared with Db + Ato5 group, ###$p < 0.001$

compared with Db + Ato10 group; **e** GHbA1c levels. *n* = 6 in each group. **f** ITT and AUC for *db/db* mice at 40 weeks. *n* = 4 per time point, and *n* = 5 per time point in Db + Ato5 group. **g** Immunohistochemical images of the RAGE in kidney sections. RAGE positive staining was mainly localized in the plasma membrane domains, and arrows represent RAGE expression in the cytoplasm of renal tubular epithelial cells[47]. Original magnification ×400. Scale bar: 50 μm. All image part of the kidney was cortex. Renal structures indicated as proximal tubule (PT). **h** Quantitative analysis of expression of RAGE. *n* = 6 in each group. Data are expressed as means ± SEM (**b**, **d**, **e**, **f**, **h**). Significance tests were two-tailed, one-way ANOVA followed by Tukey's test was performed (**b**, **d**–**f**, **h**). Source data are provided as a Source Data file.

As shown in Fig. 2n, the level of mouse urinary KIM1 in statin-treated group began to increase from the 19th week, which was consistent with the data of UACR (Fig. 2m), and serum creatinine levels were also significantly elevated compared to those in the Db group (Fig. 2k). Similarly, we observed a significant increase in serum creatinine levels in STZ-induced diabetic mice (Supplementary Table 3). The GFR is a more direct reflection of glomerular injury, and the results indicated that long-term statins administration led to a significant decrease in the GFR in *db/db* mice (Fig. 2l). These data suggested that long-term statins administration worsens renal injury in diabetic mice. In our study, we also used metformin in combination with atorvastatin and found that this treatment significantly reduced renal injury caused by statins (Supplementary Fig. 8).

## Long-term statins administration worsens renal fibrosis in *db/db* mice

To explore the causes of long-term statin-induced renal injury, we further examined renal fibrosis. In addition to Masson staining (Fig. 3a, e) and Sirius red staining (Fig. 3b), we also measured the expression of collagen, type I (COL1A1) (Fig. 3c, f) and α-smooth muscle actin (α-SMA) (Fig. 3d, g). These data further confirmed that long-term statins administration exacerbates renal fibrosis, which leads to renal injury.

## Long-term statins administration worsens renal inflammation and renal tubular epithelial cell apoptosis in *db/db* mice

Inflammation is a key factor in fibroblast activation in the kidneys of diabetic mice, and fibrosis or persistent inflammation contributes to the development of DN. First, we used an immunohistochemical assay to examine the macrophage marker CD68 to evaluate the proinflammatory effects of long-term statins administration on the diabetic kidney. Figure 4a shows that macrophage infiltration of renal tissues was extensive in the statins administration groups. The activation of NF-κB signaling eventually leads to the activation of macrophages[22]; therefore, we used an immunofluorescence assay to evaluate the involvement of NF-κB signaling. Figure 4b shows that long-term statins administration significantly activated NF-κB entry into the nucleus, as expected. IL-1β is downstream of NF-κB and is mainly synthesized and released by infiltrating macrophages in the kidney[23]. The inflammatory outbreak led to apoptosis in renal tubular epithelial cells, as shown by TUNEL staining (Fig. 4e). We examined the protein expression levels of IL-1β and NGAL, and the immunoblot results demonstrated that long-term statins administration markedly increased IL-1β and NGAL expression in the kidneys of *db/db* mice (Fig. 4f–h). Immunohistochemical staining further confirmed the immunoblot findings (Fig. 4c, d). Overall, these data indicate that long-term statins administration worsens renal inflammation and tubular epithelial cell apoptosis and contributes to accelerating the development of DN.

## Long-term statins administration contributes to lipid accumulation in *db/db* mice

To explore how long-term statins administration causes renal injury in *db/db* mice, we used RNA-Seq analysis to analyze the kidneys in the Db

group and the Db + Ato10 group (Fig. 5a). RNA-Seq showed that genes were significantly enriched in lipid metabolism processes, compared to those in the Db group. We subsequently measured renal TG levels and found that the levels were significantly higher in the statins administration group than in the Db group (Fig. 5b). Oil Red O staining showed that statins administration exacerbated lipid droplets (LDs) deposition compared with that in the Db group (Fig. 5d). Furthermore, increased LDs deposition was observed in tubular epithelial cells by TEM (Fig. 5e), and this finding was consistent with the oil red O staining results. The FFAs uptake fluorescence assay showed that long-term statins administration increased FFAs uptake in the kidney in *db/db* mice, which could lead to lipid deposition (Fig. 5c). 4-HNE, a toxic end product of lipid peroxidation, accumulates in the kidney and can exacerbate renal injury[24]. Immunohistochemical staining of 4-HNE and DHE staining suggested that a large amount of LDs deposition led to reactive oxygen species (ROS) overproduction, further leading to overproductions of the lipid peroxidative product 4-HNE, which may activate the inflammatory response in the kidney, leading to renal fibrosis (Fig. 5f, g).

However, long-term statins administration significantly reduced LDs deposition in the livers of *db/db* mice (Supplementary Fig. 9a, b). These data suggest that statins exacerbate diabetic renal LDs deposition independent of their CHOL-lowering effect.

## Long-term statins administration has a CHOL-lowering effect but increases renal lipid uptake and inhibits FAO in *db/db* mice

To further explore renal lipid deposition, we first examined the serum lipid profile. Serum TCHO, TG, and LDL levels in the statins administration group were significantly lower than those in the Db group (Fig. 6a). STZ-induced diabetic mice also exhibited the same lipid-lowering effect (Supplementary Table 3). We further examined the CHOL-lowering targets of statins, HMGCR and low-density lipoprotein receptor (LDLR) and found that statins significantly inhibited HMGCR and upregulated LDLR expression in the liver (Supplementary Fig. 9c–g). In contrast, statins upregulated the expression levels of HMGCR, LDLR and the transmembrane protein CD36 (also known as scavenger receptor B2) in the kidney, and consistent results were obtained by immunohistochemistry and immunoblotting (Fig. 6b–h), suggesting that increased renal lipid uptake may be associated with statin-induced LDs deposition. To verify that statin-induced LDs deposition was associated with LDLR upregulation, we used an STZ-induced T2DM model in *LDLR*$^{-/-}$ mice and found a significant reduction in lipids after long-term statins administration (see below). We further found a significant reduction in LDs deposition in the kidneys by nile red staining (see below), suggesting that the upregulation of LDLR may be associated with statin-induced LDs deposition in the kidney.

Adipose triglyceride lipase (ATGL) has been reported to serve as a rate-limiting enzyme in lipolysis, and ATGL-deficient mice exhibit albuminuria accompanied by ectopic deposition of fat in the kidney[25]. Carnitine palmitoyl transferase 1α (CPT1) is a rate-limiting enzyme in FAO, and inhibiting FAO in tubular epithelial cells cause ATP depletion, cell death, dedifferentiation and intracellular LDs deposition[26]. In our study, long-term statins administration

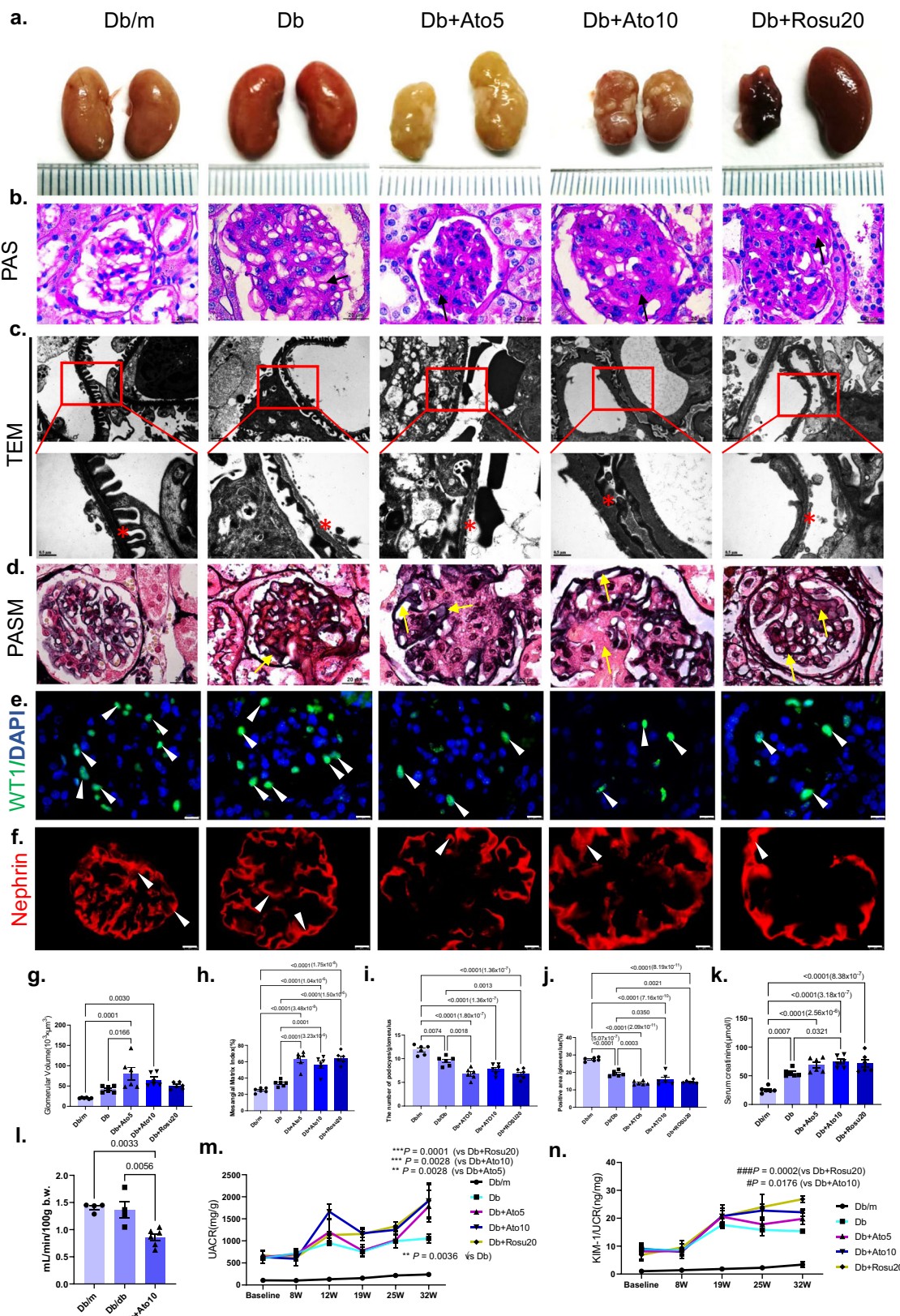

significantly reduced renal ATGL and CPT1α expression (Fig. 6e, i, j), which further accelerated LDs deposition. Overall, these data indicate that although long-term statins administration significantly reduces blood lipid levels, it also activates CHOL synthesis and endocytosis in the kidney and further inhibits FAO and lipolysis, leading to LDs deposition in the kidney.

**Long-term statins administration activates sterol-regulatory element binding protein-1 (SREBP-1) through the PI3K/Akt/mTOR pathway in *db/db* mice**

The level of lipids in kidney tissue depends not only on the plasma level and uptake capacity but also on kidney lipid synthesis and metabolism. As mentioned previously, long-term statins administration can lead to

**Fig. 2 | Long-term statins administration worsens renal injury in *db/db* mice.** All mice were ~50 weeks old. **a** Representative images of mice kidney. **b** Representative sections of PAS staining for glomerulus. Arrows represent mesangial expansion and thickened capillary loop. Original magnification ×1000, Scale bar: 20 μm. **c** Representative images of TEM. The ultrastructure of renal cortex under electron microscope was shown in the enlarged pictures, and the stars in pictures were used to marked the basement membrane thickening and foot process fusion. Original magnification ×9700 and ×26,500, scale bar: 1 and 0.5 μm. **d** Representative sections of Periodic Acid-Silver Methenamine (PASM) staining for glomerulus. Arrows represent histologic abnormalities in the glomerulus, such as the increase of mesangium and GBM expansion, in the statin administration groups. Original magnification ×1000, Scale bar: 20 μm. **e** Immunofluorescence of Wilm tumor gene1 (WT1) staining. FITC-labeled WT1 (green) and DAPI (nuclei, blue) were used. WT1 is mainly expressed on the nucleus of glomerular podocytes, and the specific positive staining is indicated by the arrows[48]. Original magnification ×1000, Scale bar: 10 μm. **f** Immunofluorescence of Nephrin staining. Alexa Fluor 594-labeled Nephrin were used. Nephrin is mainly expressed on the plasma membrane domains, and arrows represent Nephrin positive staining[49]. Original magnification ×1000, Scale bar: 10 μm. **g** Glomerular Volume ($10^3 \times \mu m^3$). **h** Quantification of the mesangial area glomerulus. **i** Quantification of the number of podocytes in glomerulus. $n = 6$ in each group. **j** Immunofluorescence analysis of Nephrin. **k** Creatinine. **l** GFR measurement. $n = 4$ in Db/m and Db/db group, $n = 6$ in Db + Ato10 group. **m** UACR ($n = 4$ per time point). Data are expressed as means ± SEM. Significance \*\*$p < 0.01$ versus Db/m group; \*\*\*$p < 0.001$ versus Db/m group. **n** Urinary KIM1 levels ($n = 4$ per time point). Significance #$p < 0.01$ versus Db group, ###$p < 0.001$ versus Db group. $n = 6$ in each group (**g**–**k**). Data are expressed as means ± SEM (**g**–**l**). One-way ANOVA with Tukey post hoc test was used for the analysis of statistical significance. Source data are provided as a Source Data file.

LDs deposition through increased lipid uptake and decreased FAO. Thus, we hypothesized that long-term statins administration activates SREBP-1-mediated fatty acid synthesis, which is involved in LD deposition. It has been reported that SREBP-1 can promote the expression of fatty acid synthase (FAS) and the fibrosis-promoting factor transforming growth factor β, which in turn promotes kidney lipid deposition and extracellular matrix accumulation[27,28]. As shown by immunohistochemistry and immunoblotting, long-term statins administration significantly increased the expression of SREBP-1 (precursor and mature form) (Fig. 7a, d, e). Similar results were shown for stearoyl-coenzyme A desaturase 1 (SCD1), acetyl-coenzyme A carboxylase 1 (ACC1) and FAS, which are downstream targets of SREBP-1, by immunohistochemistry and immunoblotting (Fig. 7b–e).

Studies have shown that SREBP-1 transcription is mainly regulated by the insulin signaling pathway and partially through PI3K/Akt/mTOR[29,30]. Next, we performed experiments to confirm whether activating SREBP-1 was dependent on the PI3K/Akt/mTOR pathway. As shown by immunoblot analysis, long-term statins administration significantly increased PI3K phosphorylation and activated its downstream kinase activity, including that of Akt and p70s6k (Fig. 7d, e). To further verify that SREBP-1 was activated through the PI3K/Akt/mTOR pathway in the kidney, we used HK-2 cells for in vitro validation. First, the cells were treated with 2.4 μM insulin for 24 h, followed by 120 μM oleic acid (OA) for 24 h, and SREBP-1 was significantly activated through the PI3K/Akt/mTOR pathway. In addition, we used the Akt1/2/3 inhibitor MK-2206 2HCl (1 and 10 μM) and the mTOR inhibitor CCI-779 (at 1 and 10 μM) for 24 h. Activation of the downstream kinases Akt and p70s6k was significantly abolished by the inhibitors (Fig. 8b, c). BODIPY staining showed that insulin-treated HK-2 cells exhibited exacerbated OA-induced intracellular LD deposition, accompanied by significant activation of SREBP-1 (Fig. 8a I and II). In contrast, LDs deposition was significantly reduced by the addition of inhibitors, while the activation of SREBP-1 was inhibited (Fig. 8a III and IV). These data suggest that LDs deposition in the kidney is modulated by SREBP-1 activation through the PI3K/Akt/mTOR pathway.

To verify that statin-induced LDs deposition is associated with SREBP-1 upregulation, we used an STZ-induced T2DM model in *srebp1*-deficient mice and found a significant reduction in lipids after long-term statins administration (Fig. 9a). PAS and Masson staining revealed that long-term administration of statins in diabetic mice after SREBP-1 knock down showed significant renoprotective effects (Fig. 9c, d, f, g). Compared with the WT group, the glomerular mesangial of the mice in the STZ group had a significant expansion, and renal fibrosis was significantly aggravated, and that in the STZ + ATO10 group was even more pronounced. In contrast, the glomerular mesangial expansion of STZ + ATO10 group in LDLR$^{-/-}$ and srebp1-deficient mice was not significantly changed compared with STZ group (Fig. 9c, d). We further found a significant reduction in LDs deposition in the kidneys by nile red staining, suggesting that SREBP-1

upregulation may be associated with statin-induced LDs deposition and renal injury (Fig. 9b, e).

Moreover, we treated HK-2 cells with OA (120 μM) for 24 h to simulate the diabetic state, and treated the cells with or without statins (Supplementary Fig. 10a, b). However, it could not have the same effect on increased SREBP-1 as significantly as when stimulated with insulin. It is difficult to simulate the effects of animal experiments by directly stimulating HK-2 cells with statins. Therefore, we used a cellular model of insulin resistance to simulate the effects of statin-induced insulin resistance on HK-2 cells.

## Discussion

Statins are a class of drugs that reduce serum LDL CHOL levels by acting as potent, specific, competitive inhibitors of HMGCR, which is a microsomal enzyme that catalyzes the conversion of HMG-CoA to mevalonate during the rate-determining step of CHOL metabolism[31]. Furthermore, statins can upregulate LDLR expression in the liver and peripheral tissues. Numerous evidence-based studies have demonstrated the safety and benefits of statins therapy in significantly reducing cardiovascular events and these drugs are recommended for the management of DN. However, in recent years, the side effects of statins have attracted increasing attention, especially the increased risk incidence of new-onset T2DM. Recently, a retrospective matched-cohort study showed that statins use was associated with diabetes progression, and suggested that the risk-benefit ratio of statins use in patients with diabetes should take into consideration its metabolic effects[16]. Therefore, in DN patients, long-term statins administration, which is recommended by the guide, may amplify these side effects, increase insulin resistance, interfere with glucose metabolism, worsen blood glucose control, and ultimately accelerate the progression of DN. In the current study, we initially confirmed the effect of long-term statins administration on DN, which extends our knowledge of statins and their disturbance of lipid metabolism in the diabetic kidney. These findings are completely opposite to the effects of statins in nonalcoholic fatty liver (NAFLD), in which they reduce hepatic lipid deposition. However, in diabetic kidney, long-term statins administration markedly increases lipid deposition, leading to EFD. Based on previous studies showing that statins treatment induces insulin resistance, and our results confirming that statins treatment exacerbates insulin resistance under diabetic conditions, we hypothesize that worsened insulin resistance leads to disturbed kidney lipid metabolism and exacerbates the progression of DN.

As evidence continues to accumulate demonstrating the link between disturbances in insulin resistance and statins, these insights indicate potential therapeutic side effects for diabetes[32]. Functionally, we demonstrated that long-term statins administration exacerbates insulin resistance, resulting in significantly increased blood glucose and aggravation of inflammation and fibrosis. Long-term statins administration resulted in markedly altered lipid metabolism, as shown by RNA-seq analysis. The data demonstrated that long-term

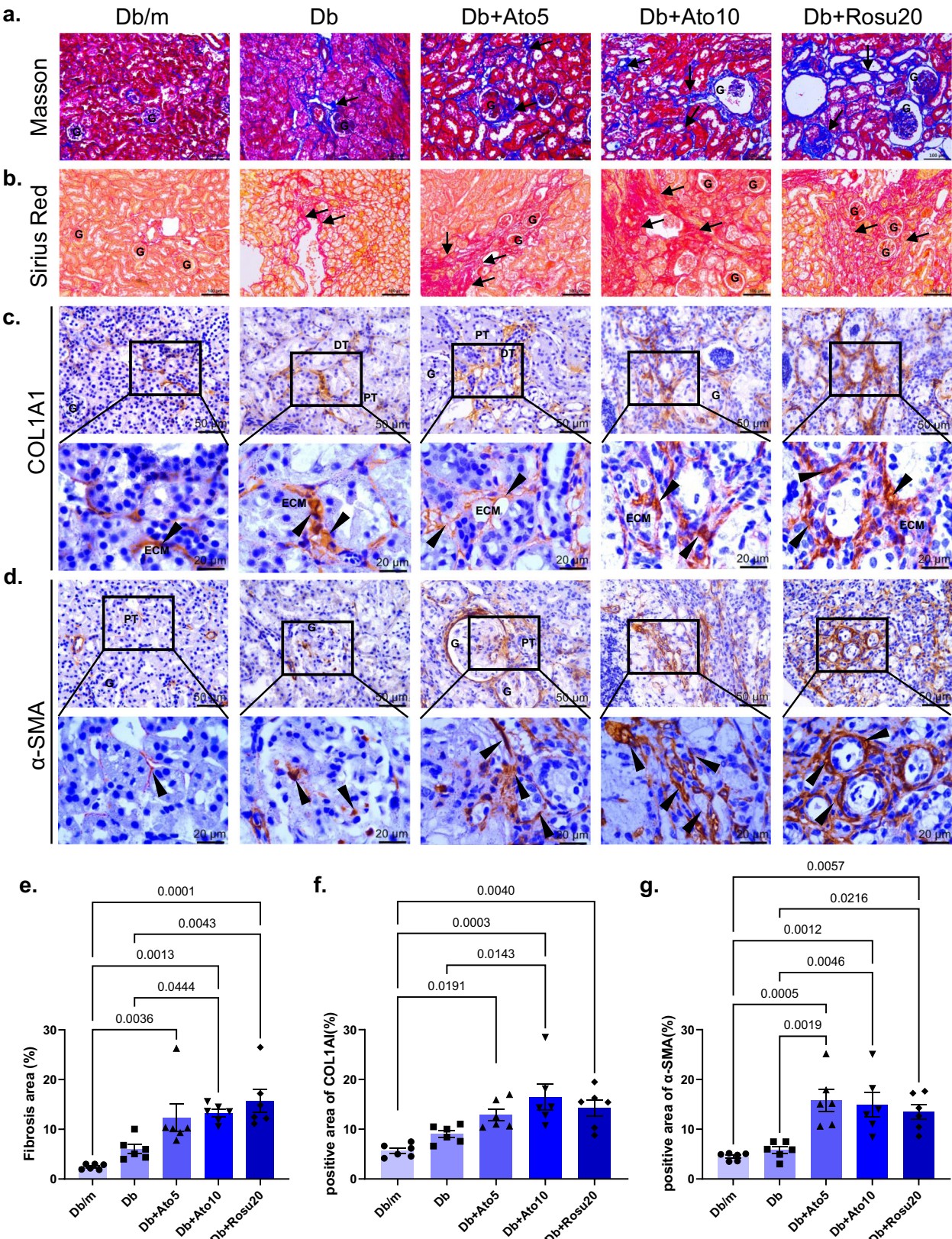

statins administration leads to lipid deposition and exacerbates kidney injury. These results indicate that statins may play different roles in the regulation of lipid metabolism in different tissues under different pathological conditions.

Why do statins, which are lipid-lowering drugs, reduce blood CHOL but also induce renal lipid accumulation? We examined HMGCR, a target of statins, and found that mice that long-term statins administration significantly upregulated renal HMGCR expression in mice, which we believe is associated with statin resistance. In recent years, statin resistance has received increasing attention[33]. Ruan et al. demonstrated that inflammatory stress induced statin resistance and increased intracellular CHOL synthesis by enhancing HMG-CoA-R

**Fig. 3 | Long-term statins administration worsens renal fibrosis in *db/db* mice.** All mice were ~50 weeks old. **a** Representative Masson's trichrome staining. In the picture, collagenous components are stained as blue color and cytoplasm is varying shades of red. Collagen deposits (blue) are evident within the fibrotic interstitial lesions between tubular, and even in the glomerulus, as marked by arrows. Original magnification ×200. Scale bar: 100 μm. **b** Representative images of Sirius red staining for tubulointerstitial fibrosis. Sirius red staining showed fibrosis (Red color) in kidney fibrotic interstitial lesions, as marked by arrows. Original magnification ×200. Scale bar: 100 μm. **c** Immunohistochemistry of COL1A1 staining. COL1A1 is mainly expressed on the extracellular matrix (ECM), and the specific location is indicated by the arrows[50]. Original magnification ×400 or ×1000. Scale bar: 100 or 20 μm. **d** Immunohistochemistry of α-SMA staining. α-SMA is used as a marker for a subset of activated fibrogenic cells, myofibroblasts, which are regarded as important effector cells of tissue fibrogenesis. Arrows represent numerous activated fibrogenic cells in tubulointerstitium and glomeruli[50]. Original magnification ×400 or ×1000. Scale bar: 100 or 20 μm. **e** Quantification of tubulointerstitial fibrosis in the kidney cortex. **f, g** Quantification of immunohistochemical staining. All image part of the kidney was cortex. Renal structures indicated as glomerulus (G), proximal tubule (PT), and distal tubule (DT), extracellular matrix (ECM). Data are expressed as means ± SEM. $n = 6$ in each group. One-way ANOVA with Tukey post hoc test was used for the analysis of statistical significance. Source data are provided as a Source Data file.

activity[34]. He further confirmed that inflammatory stress increased intracellular CHOL synthesis by disrupting SCAP-mediated HMG-CoA-R feedback regulation in the kidney, which causes "kidney statin resistance"[35]. In our study, we found severe renal inflammation in mice, which may lead to the onset and progression of statin resistance. Statin resistance leads to lipid accumulation and further exacerbates inflammation, forming a vicious cycle, that ultimately leads to EFD.

Statins not only inhibit cellular CHOL synthesis but also upregulate LDLR expression in the liver and peripheral tissues, leading to increased removal of LDL-C from the blood. However, the upregulation of LDLR in the kidney may result in increased LDLR-mediated uptake of LDL-C in the kidney. These findings suggest that the kidney is impaired by lipid accumulation mediated by the upregulation of LDLR expression and subsequent increase in LDLR-mediated uptake of LDL-C. In addition, we observed that the CD36 expression in the kidneys was significantly upregulated after long-term statins administration in mice. CD36 is a multifunctional receptor that mediates the binding and cellular uptake of long-chain fatty acids and oxidized lipids and has roles in lipid accumulation, inflammatory signaling, energy reprogramming, apoptosis and kidney fibrosis[36]. Upregulated CD36 expression in the renal proximal tubule indicates increases in CD36 retrieve albumin-bound fatty acids and oxidized lipids from the filtrate[37]. This change results in the accumulation of intracellular lipids and leads to accelerated progression of DN. Overall, we hypothesize that diabetic patients with inflammation may experience secondary distribution of CHOL after long-term statins administration, which reduces serum CHOL levels and increases kidney lipids due to increased HMGCR activity in the kidney and the upregulation of LDLR and CD36 expression.

Kidney lipid metabolism includes fatty acid synthesis, fatty acid uptake and FAO. In addition to fatty acid uptake, we found that fatty acid synthesis, the SREBP pathway, and other FAO-related genes were involved. SREBPs are major transcription factors that regulate the biosynthesis of fatty acids, CHOL, and TG. In humans, there are two SREBP genes, SREBP-1 and SREBP-2. SREBP-1 induces genes encoding enzymes that catalyze various steps in the fatty acid and TG synthesis pathways, such as ACC1, FAS, and SCD1[38]. Our results suggest that long-term statins administration increases in SREBP-1 and downstream FAS, SCD1, and ACC1 levels in the kidneys of diabetic mice, increases HMGCR expression. How do statins stimulate SREBP-1 activation by nuclear translocation? It is well known that insulin can potently induce de novo lipogenesis by selectively stimulating the processing of SREBP-1[39]. Accumulating evidence has shown that statins impair β-cell function, decrease insulin sensitivity and increase insulin resistance, which are related to an increase in new-onset diabetes mellitus among patients receiving statins[40,41]. In our study, insulin concentrations were significantly increased in the statin groups. Several lines of evidence suggest that the insulin-mediated PI3K/Akt/mTOR signaling pathway can regulate lipid biosynthesis through SREBP-1 and related lipogenic enzymes[42]. Therefore, we hypothesized that long-term statins administration exacerbates DN through the insulin-mediated PI3K/Akt/mTOR pathway. Our study showed that long-term statins administration activated the insulin-mediated PI3K/Akt/mTOR pathway and

further activated the expression of the major downstream effectors FAS, ACC1, and SCD1, ultimately increasing intracellular lipid accumulation. The activation of SREBP-1 signaling in glomeruli correlated with lipid accumulation and the GFR in subjects with podocytopathies, including DN[43]. Furthermore, Gabitova-Cornell et al. showed that disrupting CHOL biosynthesis with statins switched glandular pancreatic carcinomas to a basal subtype via the activation of SREBP-1[18], which is consistent with our results. In the present study, we investigated several FAO-related genes, such as ATGL and CPT1α. Our results showed the downregulation of renal ATGL and CPT-1α expression in mice after long-term statin administration, suggesting that FAO and lipolysis were inhibited, leading to LD deposition in the kidney. Our findings indicate that long-term statins administration may promote fatty acid intake and reduce FAO and TG decomposition, which lead to EFD in the kidney. EFD in renal epithelial tubular cells could induce the production of ROS and subsequently increase the expression of inflammatory and fibrotic factors, which was associated with the progression of DN[4].

Furthermore, we used diabetic LDLR$^{-/-}$ mice and diabetic *srebp1*-deficient mice to validate the mechanism by which long-term statins administration leads to renal EFD in diabetic mice. We observed amelioration of renal EFD and tubular injury in both diabetic LDLR$^{-/-}$ mice and diabetic *srebp1*-deficient mice with DN, suggesting that long-term statins administration may induce renal EFD through the upregulation of renal LDLR and SREBP-1 expression. In the future, additional experiments in which long-term statins administration exacerbates DN in more robust animal models with more features of human DN may be performed.

No drugs are entirely free of adverse effects, and statins are no exception. Overestimating the clinical benefits of a drug or underestimating the risk will lead to public health problems. Although increased statin-induced insulin resistance has attracted much attention in recent years, we still need to recognize that the powerful CHOL-lowering and cardioprotective effects of statins vastly outweigh the potential increase in the risk of insulin resistance. Our results provide a new focus for long-term statins administration in the treatment of hyperlipidemia in patients with T2DM, revealing the risk of related renal function decline. Statins users should be monitored regularly for changes in glycemia and GHbA1c levels, and glucose-lowering medication must be adjusted in a timely manner to reduce the risk associated with long-term statins administration. In our study, we used metformin in combination with atorvastatin and found that this treatment significantly reduced renal injury caused by statins. Therefore, the combination of multiple glucose-lowering medications in patients with diabetes can alleviate the side effects of statins. One of the biggest limitations of our study was the inability to conduct a prospective clinical study to validate our findings. However, our findings in animal studies are consistent with a retrospective study by Mansi et al. that showed that statins use was associated with diabetes progression in patients with diabetes, which included data on statins use in patients with diabetes for up to 12 years[16]. The risk-benefit ratio of statins use in patients with diabetes should take into consideration the metabolic effects. Understanding

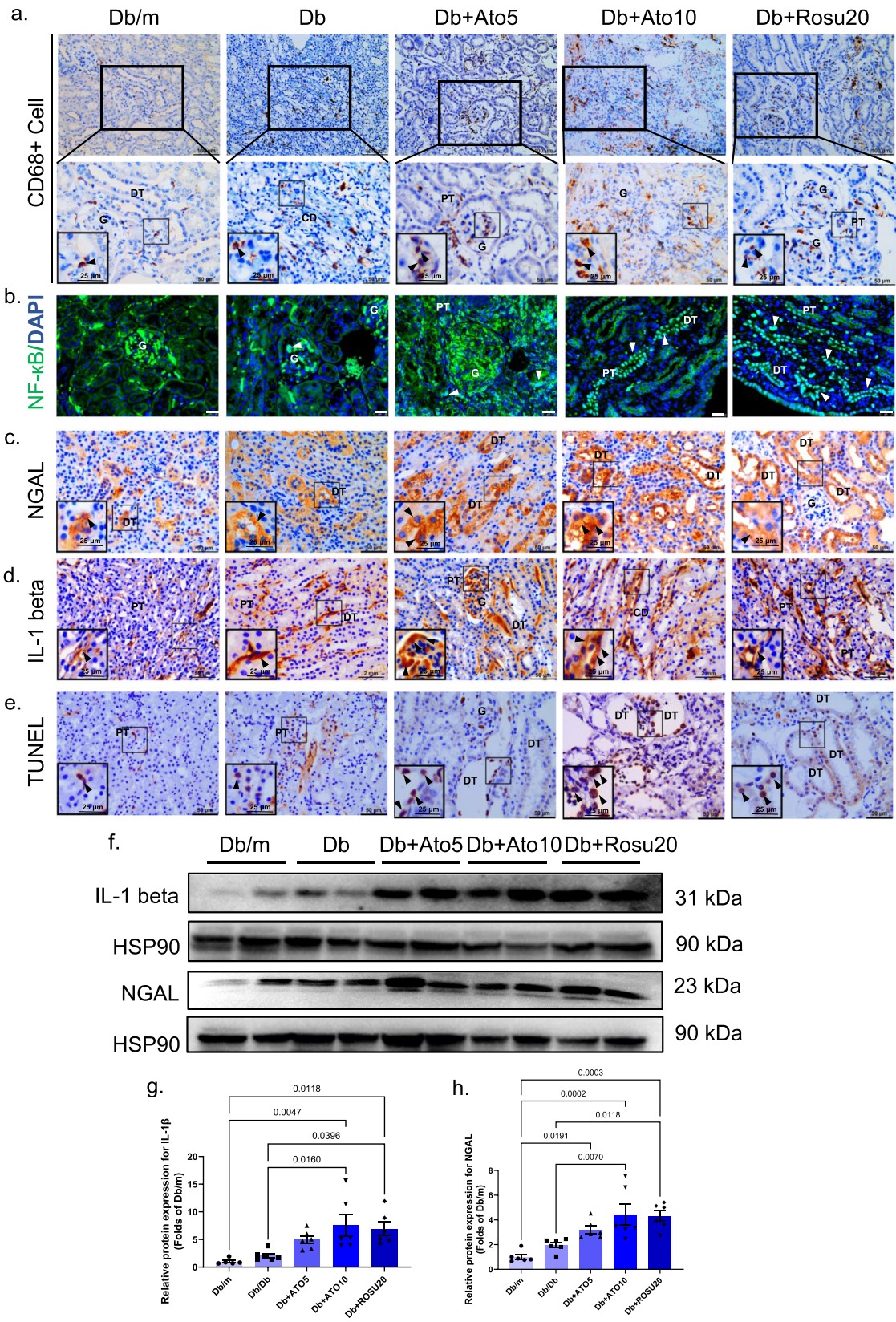

the mechanism underlying this phenomenon could lead to the development of treatments to balance the cardiovascular benefits of statins therapy with the risk of diabetes progression. In future studies, it would be interesting to further examine a therapeutic regimen that has the potential to become a common addition to statins treatment in T2DM patients to reduce risk.

Collectively, our study shows that long-term statins administration has a CHOL-lowering effect and reduces NAFLD, but it also exacerbates insulin resistance, induces renal EFD and impairs renal function (Fig. 10). Our study suggests that T2DM patients should be monitored regularly for changes in blood glucose and GHbA1c levels to reduce the risk associated with long-term statins administration.

**Fig. 4 | Long-term statins administration worsens renal inflammation and tubular cell apoptosis in *db/db* mice.** All mice were -50 weeks old.
**a** Immunohistochemistry of CD68[+] cell staining. CD68 is a valuable marker which can be used to identify macrophages and monocytes. And the arrows represent numerous macrophages in tubulointerstitium and glomeruli[51]. Original magnification ×200 and ×1000. Scale bar: 100 and 20 μm. **b** Immunofluorescence of NF-κB staining. FITC-label (green) NF-κB and DAPI (nuclei, blue) were used[52]. White arrows mean NF-κB-positive nuclei. Original magnification ×400. Scale bar: 25 μm.
**c** Immunohistochemistry of neutrophil gelatinase-associated lipocalin (NGAL) staining. NGAL is mainly expressed in the cytoplasm, and the specific location is indicated by the arrows[53]. Original magnification ×400. Scale bar: 50 μm.

**d** Immunohistochemistry of IL-1β staining. IL-1β is mainly expressed in the cytoplasm, and the specific location is indicated by the arrows[54]. Original magnification ×400. Scale bar: 50 μm. **e** Representative sections of TUNEL-positive cells. TUNEL-positive cells are expressed in nucleus, and the specific location is indicated by the arrows. Original magnification ×400. Scale bar: 50 μm. **f** The immunoblot analysis of IL-1β and NGAL. **g, h** Analysis of the grayscale image between them. All image part of the kidney was cortex. Renal structures indicated as glomerulus (G), proximal tubule (PT), and distal tubule (DT), collection tube (CD). Data are expressed as means ± SEM. *n* = 6 in each group. One-way ANOVA with Tukey post hoc test was used for the analysis of statistical significance. Source data are provided as a Source Data file.

## Methods
### Ethics statement
All experiments performed in this study were compiled with ethical regulations and approved by the Animal Care and Ethics Committee of Sun Yat-sen University. The details are described in the respective sections below.

### Animal model and treatment
*Db/m* (C57BLKS/J background) mice and *db/db* mice were purchased from GemPharmatech Co., Ltd (Jiangsu, China) following breeding at the Center for Disease Model Animals of Sun Yat-sen University. The *db/m* is the heterozygous mouse without diabetes, and are often used as the control for *db/db* mice. Ten-weeks-old male *db/db* mice and *db/m* mice were enrolled in these experiments and divided into five groups: Db/m group, Db group, Db + Ato5 group (atorvastatin 5 mg kg⁻¹ BW/day), Db + Ato10 group (atorvastatin 10 mg kg⁻¹ BW/day) and Db + Rosu20 group (Rosuvastatin 20 mg kg⁻¹ BW/day). Mice were randomly assigned to experimental groups. Only male mice were enrolled. *n* = 16 in each group. In addition, we also used *db/m* mice as a control group administered statins and divided into four groups: Db/m + Ato5 group, Db/m + Ato10 group; Db/m + Rosu20 group. *n* = 8 in each group. In addition, we set up two groups, the Db + Met group (metformin 100 mg kg⁻¹ BW/day) and the Db + Met + Ato10 group (metformin 100 mg kg⁻¹ BW/day + atorvastatin 10 mg kg⁻¹ BW/day). *n* = 6 in each group. The dose of Atorvastatin and Rosuvastatin was determined based on previous studies[17–19]. Statins were suspended in 0.9% saline and administered five consecutive days a week via oral gavage. All animals were given water and chow diet (purchased by Guangdong Medical Laboratory Animal Center, consisting of fat [4.8%], protein [18.6%], and carbohydrate [61%]) during the whole experiment period. During this period, body weight, FBG were measured. The duration of our observation lasted until the 50th weeks. Briefly, the mice were anesthetized with 1% isoflurane. At the end of the experiment, kidney and liver tissues were fixed and embedded for subsequent detection. All serum and tissue samples were stored at −80 °C.

Streptozotocin (STZ)-induced T2DM mouse model were constructed using a method described previously in our paper[44]. Briefly, mice of the model group were treated with HFD at 4-weeks-old and were treated with seven consecutive intravenous injections of STZ (40 mg/kg, Sigma, St. Louis, MO) in citrate buffer (pH 4.6) at 8-weeks-old, while the control animals received chow diet and the same volume of citrate buffer. The animals of model group were given 10% sucrose/water during the period from 12 h after the first STZ injection to 12 h after the last injection and were given HFD during the whole experiment period. The animals of control group were given water and chow diet during the whole experiment period. The blood glucose level was monitored with a glucometer (One Touch Ultra Easy, Life Scan, PA, USA), on 2 weeks after the last STZ injection, and animals with blood glucose levels greater than 12 mmol/l were considered diabetic. Four-weeks-old male C57BL/6J mice were purchased from Laboratory Animal Center of Sun Yat-sen University. *n* = 3 in each group. Only male mice were enrolled. LDLR-deficient (*LDLR⁻/⁻*) mice (C57BL/6JGpt

background) were obtained from the Jackson Laboratory, and *srebp1*-deficient mice were purchased from GemPharmatech Co., Ltd (Jiangsu, China) (Strain NO. T037279). Two kinds of mice were also used for induce T2DM model. *n* = 3 in each group. Only male mice were enrolled. The experimental grouping, and statins treatment of STZ-induced diabetic mice were the same as those of *db/db* mice.

Eight-week-old male C57BL/6J and KK-Ay mice were obtained from Beijing HFK Bioscience Co. Ltd. (Beijing, China). KK-Ay mice were a kind of T2DM mouse model and allowed free access to HFD[45]. KK-Ay mice are a cross between diabetic KK and lethal yellow (Ay) mice, and carry a heterozygous mutation of the agouti gene. The severity of hyperglycemia and insulin resistance is exacerbated by the introduction of Ay allele into the KK background. The genetic background of KK mice and KK-Ay mice is the inbred mouse strain of C57BL/6J, which is always used as the control for KK mice or KK-Ay mice.C57BL/6J mice were fed regular chow and there were considered as control group. *n* = 4 in each group. Only male mice were enrolled. The experimental grouping, and statins treatment of KK-Ay diabetic mice were the same as those of *db/db* mice.

Four-week-old ApoE-deficient (ApoE⁻/⁻) mice (C57BL/6JGpt background) were also randomly assigned to 6 groups: ApoE group, HFD group, HFD + Ato3 group (atorvastatin 3 mg kg⁻¹ BW/day), HFD + Ato5 group (atorvastatin 5 mg kg⁻¹ BW/day), HFD + Ato10 group (atorvastatin 10 mg kg⁻¹ BW/day) and HFD + Rosu20 group (Rosuvastatin 20 mg kg⁻¹ BW/day). *n* = 5 in each group. To induce hyperlipidemia, ApoE⁻/⁻ mice were treated with HFD (D12492, consisting of fat [60%], protein [20%], and carbohydrate [20%]; all from Readydietech Co., Ltd. (Shenzhen, P. R. China) at 4-weeks-old during the whole experiment period. After 4-weeks HFD, the ApoE⁻/⁻ mice were orally administered with statins or vehicle once per day for 12 weeks. The HFD modeling method, experimental grouping, and statins treatment of C57BL/6J mice were the same as those of ApoE⁻/⁻ mice. The HFD model group were assigned to six groups: CON group, HFD group, HFD + Ato3 group, HFD + Ato5 group, HFD + Ato10 group and HFD + Rosu20 group. Only male mice were enrolled. *n* = 4 in each HFD model group.

All mice were maintained at the Center for Disease Model Animals of Sun Yat-sen University. All mice were housed in an animal facility with a 12 h light–dark cycle and water. Mice were housed on a 12 h light–dark cycle at 22–25 °C with 40–70% humidity and allowed free access to food and water except as noted. Mice were euthanized by intraperitoneal injection of 150 mg/kg sodium pentobarbital when they reached the experimental time endpoint or when any of the following criteria were met. (1) persistent lethargy and failure to clean hair; (2) failure to respond to physical interventions or behavioral signs of human touch, including marked inactivity, dyspnea, sunken eyes, and hunched posture; and (3) abnormal central nervous responses (convulsions, tremors, paralysis, head tilt, etc.). Animal care and handling were performed in accordance with the recommendations in the Guide for the Care and Use of Laboratory Animals of the National Institutes of Health. All animal experiments were performed according to the regulations approved by the Animal Care and Ethics Committee of Sun Yat-sen University (the protocol number is 2021000957).

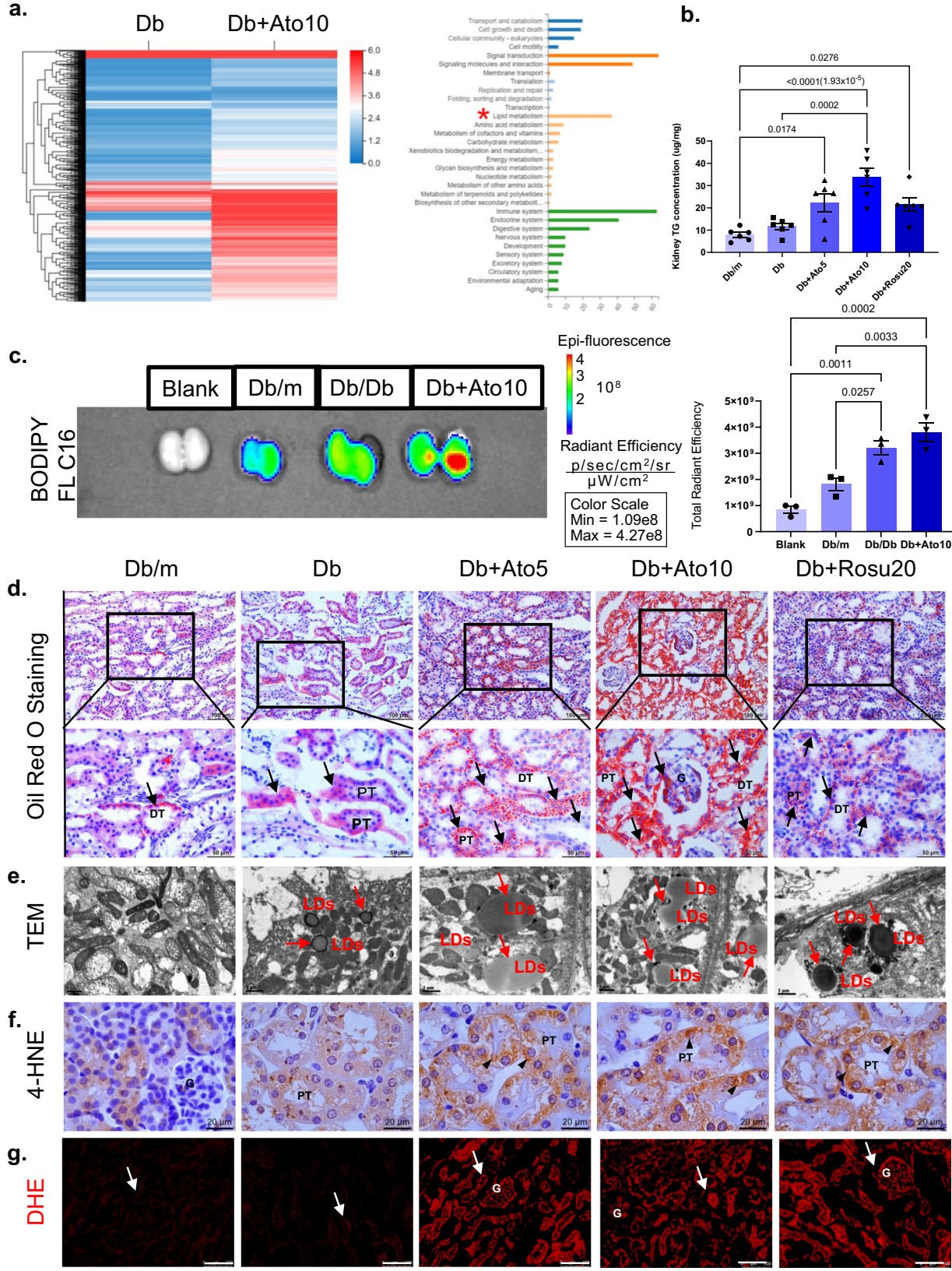

## GFR measurement

GFR was measured at 36 weeks of disease course. *Db/db* mice were injected retro-orbitally with FITC-sinistrin (7 mg 100 g⁻¹ body weight; MediBeacon, Germany). A miniaturized imager device (MediBeacon) was used to detect fluorescence in the skin on the shaved back over 1 h. GFR was calculated on the basis of the kinetics of fluorescence decay[46]. *n* = 4 in Db/m and Db/db group, *n* = 6 in Db + Ato10 group.

## Serum and urine measurement

The serum Adiponectin and Leptin levels were examined using an enzyme-linked immunosorbent assay (ELISA) (DY1119 and DY498, R&D

**Fig. 5 | Long-term statins administration contributes to lipid accumulation in *db/db* mice.** All mice were -50 weeks old. **a** Clusters of DEGs and KEGG pathway analysis of DEGs in the kidneys of Db and Db + Ato10 groups. *n* = 3 in each group. **b** The concentration of TG in kidney. *n* = 6 in each group. Data are expressed as means ± SEM. **c** Fluorescence imaging of kidney FFAs uptake from Db/m, Db and Db + Ato10 groups. *n* = 3 in each group. Data are expressed as means ± SEM. **d** Representative images of Oil Red O staining. Lipid appear as red spots, as marked by arrows. Original magnification ×200 and ×400. Scale bar: 100 or 50 μm. **e** Representative images of TEM for LDs. Original magnification ×9700, scale bar: 1 μm. *n* = 5 images from three mice per group. **f** Immunohistochemistry of

4-hydroxynonenal (4-HNE) staining. 4-HNE is mainly expressed on the cytoplasmic, and the specific location is indicated by the arrows[55]. Original magnification ×1000. Scale bar: 20 μm. **g** Immunofluorescence of DHE staining. DHE staining was used to estimate superoxide generation. Arrows represent DHE distribution was markedly increased in the statin administration groups compared to the Db group. Original magnification ×200. Scale bar: 100 μm. *n* = 10 images from six mice per group (**d**, **f**, **g**). All image part of the kidney was cortex. Renal structures indicated as glomerulus (G), proximal tubule (PT), and distal tubule (DT). One-way ANOVA with Tukey post hoc test was used for the analysis of statistical significance. Source data are provided as a Source Data file.

Systems, Minnesota, USA). The serum fasting insulin levels and Glycated Hemoglobin A1c (GHbA1c) levels were examined using an ELISA kit (E08141m, Cusabio, Wuhan, China). The serum creatinine, TG, TCHO, LDLC and HDLC levels were examined using commercial reagent kits (C011-2-1, A110-1-1, A111-2-1, A113-1-1, A112-1-1, Jiancheng, Nanjing, Jiangsu, China). Urinary hemoglobinuria test was used a Urine Hemoglobin Qualitative Detection Kit (ADS043TC0, MEIMIAN, Jiangsu, China).

### UACR and urinary kidney injury molecule-1 (KIM-1) levels
To assess urinary albumin excretion, *db/db* mice urine samples were collected by metabolic cage in 12 h per 4 weeks and stored at −80 °C. The urinary albumin was quantified using a Mouse Albumin ELISA Quantification Kit (E-EL-M0792c, Elabscience, Wuhan, China), Mouse Kidney injury molecule 1 (KIM−1) ELISA Kit (CSB-E08809m, Elabscience, Wuhan, China) and urine creatinine (C011-2−1, Jiancheng, Nanjing, Jiangsu, China) according to the manufacturer's instructions. *n* = 4 in each group.

### Insulin tolerance test (ITT)
ITT was measured at 40 weeks of disease course. To perform the ITT, 1 U kg$^{-1}$ of insulin (Novolin R, Novo Nordisk, Bagsvaerd, Denmark) was i.p. injected into the *db/db* mice had fasted for 12 h. The blood glucose levels were examined 0, 15, 30, 60 and 120 min after the injection. The curve of the blood glucose change level was drawn, and the area under the curve was calculated. *n* = 4 per time point, and *n* = 5 per time point in Db + Ato5 group.

### Tissues FFAs uptake fluorescence imaging
FFAs uptake was measured at 40 weeks of disease course. *Db/db* mice were injected with 1 μg g$^{-1}$ body weight BODIPY-FFA (D3821, Thermo Fisher, MA, USA) via the tail vein, 45 min later, the kidney was collected after removing fat and blood and rinsed in PBS. X-ray and fluorescence imaging were performed using a small animal living fluorescence imaging system (Xtrem, Bruker, Germany) and the mean photons were calculated. The detection parameters are as follows: excitation wavelength 460 nm, emission wavelength 535 nm and exposure time 2 s. *n* = 3 in each group.

### Histological staining
The kidney and liver tissues were fixed in 4% paraformaldehyde (pH 7.4) overnight for histological analysis, embedded in paraffin, and serially sectioned at 3.5 μm. Standard hematoxylin and eosin staining, Masson's trichrome stain, Sirus red staining and PAS staining were used. GBM thickness was determined by PASM staining. All the histological staining images were captured by a microscope (DFC700T, Leica, Germany).

### Transmission electron microscopy (TEM)
Kidney ultrastructure was examined under a transmission electron microscope (Tecnai G2 Spirit Twin +GATAN 832.10W; FEI; Czech Republic) using conventional methods. In brief, kidney tissues were fixed with 2.5% glutaraldehyde in 0.1 mol/l phosphate buffer (pH 7.4),

followed by 1% OsO4. After dehydration, thin sections were stained with uranyl acetate and lead citrate for observation, images were acquired digitally.

### Immunohistochemistry (IHC) and immunofluorescence (IF)
The kidney samples sections were blocked with 3% hydrogen peroxide and then performed at 95 °C for 10 min using citrate buffer (P0083, Beyotime, Shanghai, China), then blocking steps were carried out using the QuickBlock™ Blocking Buffer (P0260, Beyotime, Shanghai, China) according to the manufacturer's instructions. The sections were then incubated with various antibodies, including anti-α-SMA (1:200), anti-COL1A1 (1:200), anti-CD68 (1:100), anti-4-HNE (1:100), anti-NGAL (1:100), anti-IL−1β (1:100), anti-SREBP-1 (1:200), anti-FAS (1:100), anti- SCD1 (1:100), anti-CD36 (1:100), anti-LDLR (1:100), anti-RAGE (1:100), anti- NF-Kb (1:150), anti-WT1 (1:100), anti-Nephrin (1:100), anti-HMGCR (1:100), BODIPY™493/503 (1:1000). After incubated with primary antibody at 4 °C overnight, the sections incubated with secondary antibody (Beyotime, Shanghai, China) at 37 °C for 30 min. Visualization was accomplished using 3,3N-diaminobenzidine tertrahydrochloride (DAB) (P0202, Beyotime, Shanghai, China). Sections were counterstained with hematoxylin (G1080, Solarbio, Beijing, China). In the negative controls, the primary antibody was omitted and replaced with the blocking solution. All the histological staining and immune-fluorescence images were captured by a microscope (DFC700T, Leica, Germany). Image-Pro Plus version 6.0 software (Media Cybernetics, Inc., Rockville, MD, USA) was used to assess the integrated optical density (IOD) value of the IHC section. IOD values of immunohistochemistry sections were evaluated by using image pro plus version 6.0 software (Media Cybernetics, Inc., Rockville, MD, USA). The IOD of the digital image (magnification, ×400) was designated as the representative RAGE staining intensity. Since RAGE is widely expressed in kidney and not specifically expressed in a particular tubule, we selected ten randomly selected fields including distal tubules, proximal tubules and glomeruli with blind method. The IOD of each field was counted and the data were subjected to statistical analysis. Positive areas were semi-quantitatively analyzed by ImageJ software program (National Institutes of Health, Bethesda, MD, USA). Positive areas of the digital images (magnification, ×200, and Nephrin staining ×1000) were designated as representative for calculating the value of staining intensity. The positive areas from ten randomly selected fields were calculated by blind method and subjected to statistical analysis. Specifically, we quantified larger areas of the sections rather than specific tubules. As for potential background staining, the effect of background staining or non-specific staining on the research conclusion can be eliminated by setting up the control groups. And we can eliminate the influence of the background color on the positive expression analysis by setting the "threshold" of ImageJ or image pro plus version 6.0 which are efficient software. WT1 positive cells from 20 randomly selected fields of glomeruli sections were counted by blind method and subjected to statistical analysis. The antibodies are listed in "Reporting summary" and Supplementary Tables 4 and 5.

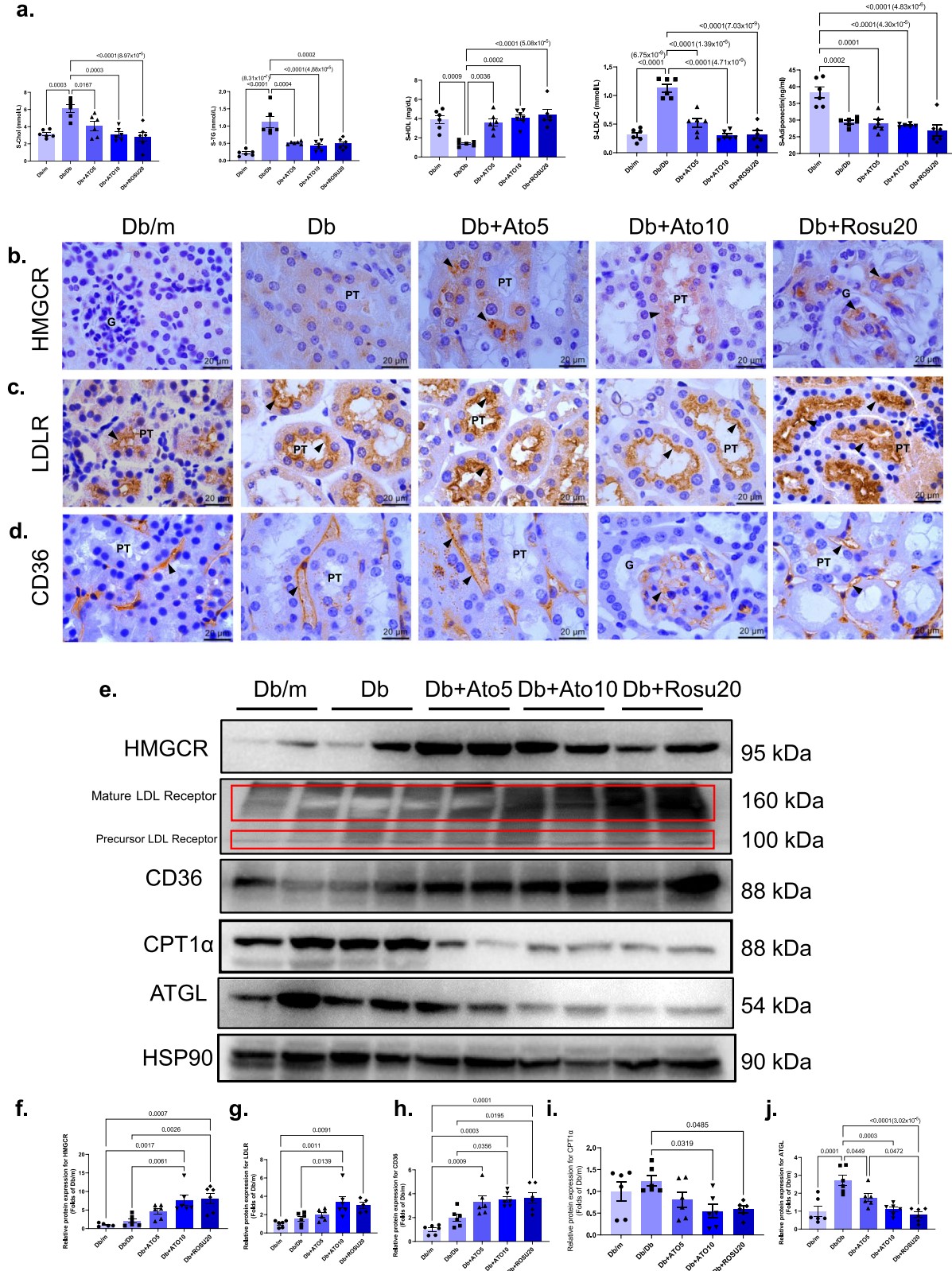

**Fig. 6 | Long-term statins administration has cholesterol-lowering effect but increased renal lipid uptake and inhibited fatty acid oxidation in *db/db* mice.** All mice were ~50 weeks old. **a** Lipid profile in *db/db* mice. *n* = 6 in each group. Data are expressed as means ± SEM. **b**–**d** Immunohistochemistry of HMGCR, LDLR, CD36 staining. HMGCR is mainly expressed on the cytoplasmic[56]; LDLR and CD36 are mainly expressed on the membrane, and the specific positive staining is indicated by the arrows[57,58]. CD36 is expressed in a wide variety of kidney cells such as proximal tubular epithelial cells, mesangial cells, podocytes, monocytes and macrophages[36]. Original magnification ×1000. Scale bar: 20 μm. **e** The immunoblot analysis of HMGCR, LDLR, CD36, CPT-1α, ATGL, and HSP90. **f**–**j** Analysis of the grayscale image between them. All image part of the kidney was cortex. Renal structures indicated as glomerulus (G), proximal tubule (PT), and distal tubule (DT). Data are expressed as means ± SEM. *n* = 6 in each group. One-way ANOVA with Tukey post hoc test was used for the analysis of statistical significance. Source data are provided as a Source Data file.

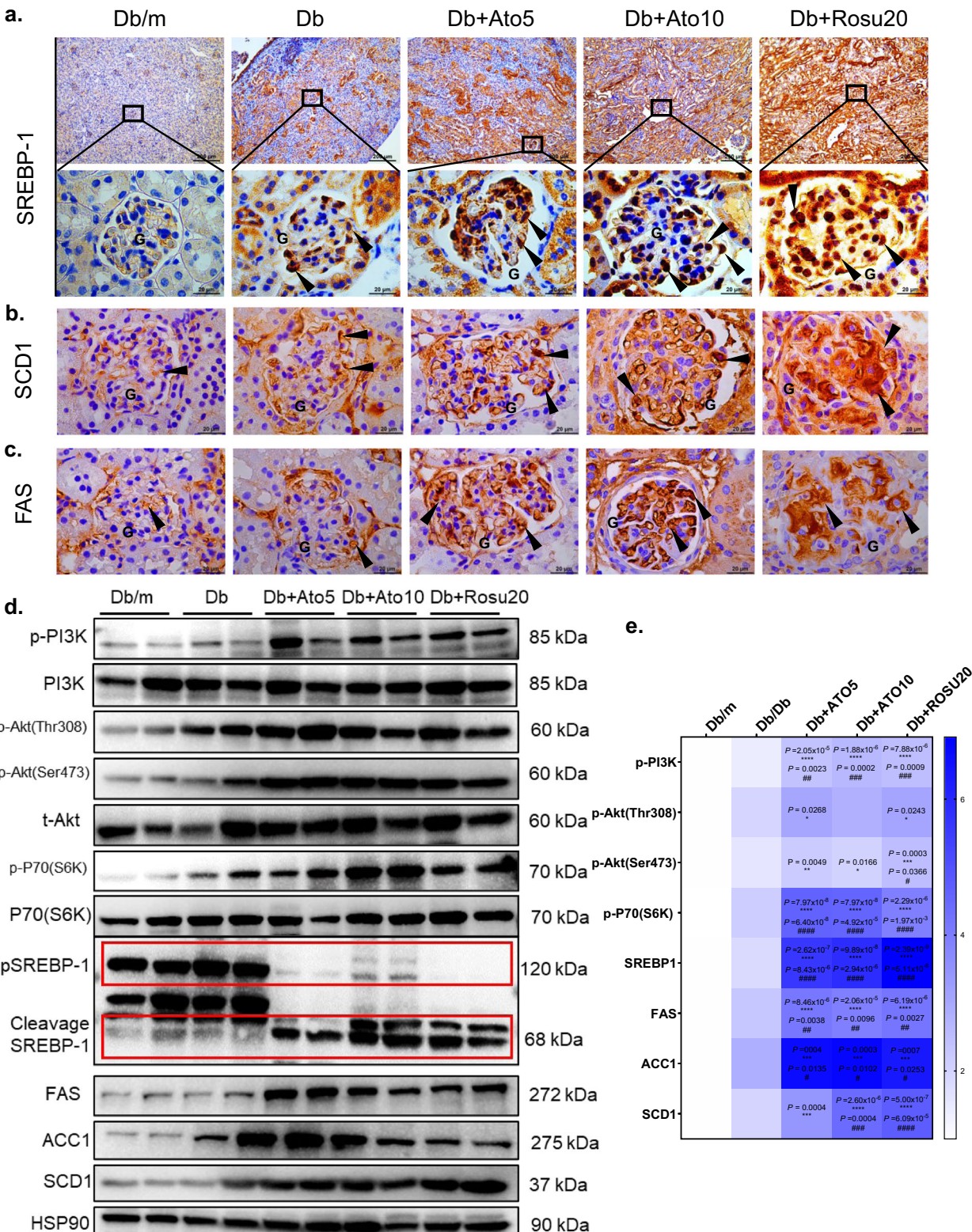

**Fig. 7 | Long-term statins administration activated SREBP-1 through PI3K/Akt/mTOR pathway in the *db/db* mice.** All mice were ~50 weeks old.
**a**–**c** Immunohistochemistry of SREBP-1, FAS, SCD1 staining. SREBP-1 is mainly expressed in the cytoplasmic and nucleus[59]; FAS, SCD1 are expressed on the cytoplasm[60,61]. Arrows in the enlargement of **a** point to positive staining for SREBP1 in the nuclei of glomerular resident cells, arrows in **b** and **c** point to positive staining for FAS and SCD1 in the cytoplasm of glomerular resident cells. Original magnification ×100 and ×1000, scale bar: 200 and 20 μm. All image part of the kidney was cortex. Renal structures indicated as glomerulus (G). **d** the immunoblot analysis of p-PI3K, PI3K, p-Akt (Thr308), p-Akt (Ser473), t-Akt, p-P70(S6K), P70(S6K), SREBP-1, FAS, SCD1, ACC1 and HSP90. **e** Analysis of the grayscale image between them. Data are expressed as means ± SEM. *n* = 6 in each group. One-way ANOVA with Tukey post hoc test was used for the analysis of statistical significance. Significance \**p* < 0.05 versus Db/m group; \*\**p* < 0.01 versus Db/m group; \*\*\**p* < 0.001 versus Db/m group, \*\*\*\**p* < 0.0001 versus Db/m group. #*p* < 0.05 versus Db group; ##*p* < 0.01 versus Db group; ###*p* < 0.001 versus Db group, ####*p* < 0.0001 versus Db group. Source data are provided as a Source Data file.

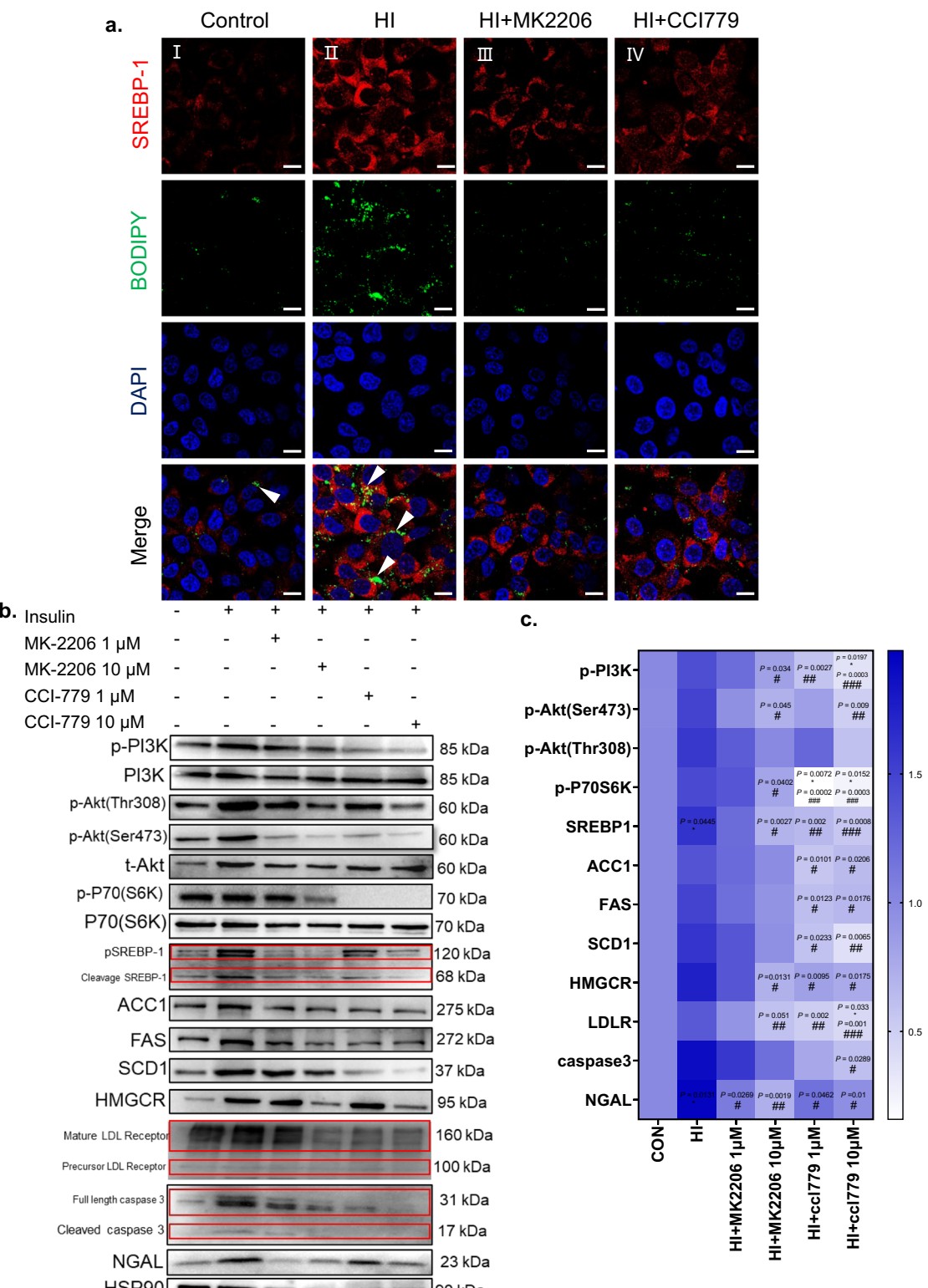

**Fig. 8 | Activating SREBP-1 in HK-2 cells aggravated lipid deposition. a** SREBP-1 (red) and BODIPY (green) expression was evaluated by immunofluorescence on HK-2 cells treated with negative control (Con) or insulin incubated with OA (120 μM) (Hight insulin, HI), or HI incubated with 10 μM Akt inhibitor MK2206 (HI + MK2206), 10 μM mTOR inhibitor CCI779 (HI + CCI779). FITC-labeled BODIPY, Alexa Fluor 594-labeled SREBP-1 and DAPI (nuclei, blue) were used. SREBP-1 is mainly expressed in the cytoplasmic of HK-2 cells, BODIPY in the cytoplasmic of HK-2 cells, and the specific location is indicated by the arrows. Representative image from three biologically independent samples/group were combined from three independent experiments. Original magnification ×630. Scale bar: 10 μm. **b** The immunoblot analysis of p-PI3K, PI3K, p-Akt (Thr308), p-Akt (Ser473), t-Akt, p-P70(S6K), P70(S6K), SREBP-1, FAS, SCD1, ACC1, HMGCR, LDLR, caspase 3, NGAL and HSP90. **c** Analysis of the grayscale image between them. Representative blots from three biologically independent samples/group were combined from three independent experiments. Data are expressed as means ± SEM. One-way ANOVA with Tukey post hoc test was used for the analysis of statistical significance. Significance *$p < 0.05$ versus CON group; **$p < 0.01$ versus CON group. #$p < 0.05$ versus HI group; ##$p < 0.01$ versus HI group; ###$p < 0.001$ versus HI group, ####$p < 0.0001$ versus HI group. Source data are provided as a Source Data file.

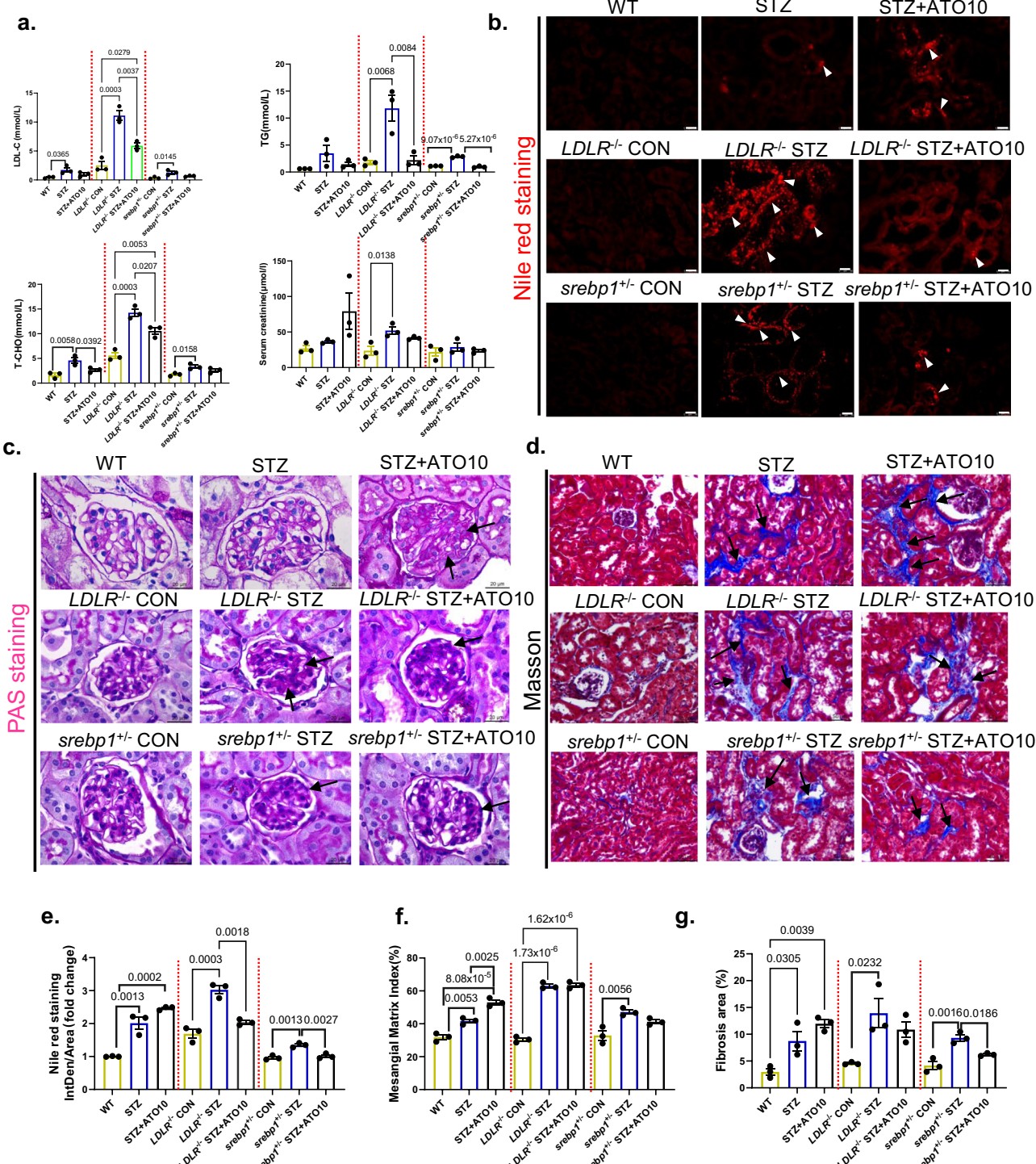

**Fig. 9 | LDLR or SREBP-1-deficient attenuates renal lipid deposition induced by long-term statins administration in STZ-induced diabetic mice.** All mice were ~30 weeks old. **a** Lipid profile and serum creatinine in STZ-induced diabetic mice. $n = 3$ in each group. Data are expressed as means ± SEM. **b** Representative images of Nile red staining. The lipid drops were stained in red, and the redder dots in the figure, the more lipid accumulation. Arrows represent lipid drops in renal tubular epithelial cells. Original magnification ×400. Scale bar: 50 μm. **c** Representative images of PAS staining. Original magnification ×1000. Scale bar: 20 μm. Arrows represent mesangial expansion. **d** Representative images of Masson's trichrome staining. In the picture, collagenous components are stained as blue color and cytoplasm is varying shades of red. Collagen deposits (blue) are evident within the fibrotic interstitial lesions between tubular, and even in the glomerulus, as marked by arrows. Original magnification ×400. Scale bar: 50 μm. **e**–**g** Quantification of **b**–**d** staining. All image part of the kidney was cortex. Data are expressed as means ± SEM. $n = 3$ in each group. One-way ANOVA with Tukey post hoc test was used for the analysis of statistical significance. Source data are provided as a Source Data file.

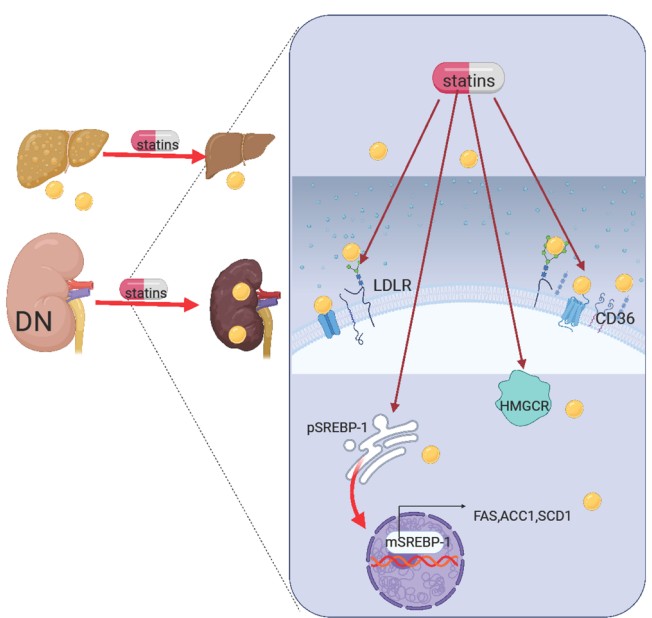

**Fig. 10 | Graphical representation of long-term statins administration exacerbates diabetic nephropathy via EFD.** Long-term administration of statins activates SREBP-1 fatty acid synthesis signaling, increases kidney lipid uptake, and inhibits fatty acid oxidation, which leads to EFD in the kidneys of diabetic mice. Graphics were created with Biorender.com.

## Oil red O staining
Frozen tissue sections (7 μm) were stained with oil red O (Sigma-Aldrich, St. Louis, MO, USA) for 20 min. All the histological staining images were captured by a microscope (DFC700T, Leica, Germany).

## TUNEL histology
Sections were deparaffinized and hydrated in xylene and gradient concentrations of ethanol, then incubated in proteinase K at room temperature for 30 min and stained with TUNEL kit (C1091, Beyotime Institute of Biotechnology, China). TUNEL assay was then performed according to the instructions by the manufacturer. All the histological staining images were captured by a microscope (DFC700T, Leica, Germany).

## Dihydroethidium (DHE) staining
We used the *db/db* mouse model for DHE staining, and the animal number of *n* was 6. Unfixed frozen kidney (7 μm) after dehydration with a 30% sucrose solution were incubated with 2 μM DHE (D7008, Molecular Probes, Sigma, USA) for 30 min in a light-protected chamber at 37 °C.

## RNA sequencing (RNA-Seq) analysis and data processing
Total RNA was extracted from kidney of *db/db* mice at 40th week of course, include Db group, Db + Ato10 group (three samples per group). The RNA library was sequenced on the BGISEQ-500 platform (BGI Genomics, Shenzhen, China). Differentially expressed genes (DEGs) were screened using two criteria: (1) a fold change greater than 1.5 and (2) a corresponding adjusted *p* value < 0.05. Kyoto Encyclopedia of Genes and Genomes (KEGG) pathway-enrichment analysis with a *p* value < 0.05 were considered significantly enriched. The analysis of DEGs and KEGG pathways was performed at Dr.tom online (https://biosys.bgi.com).

The affymetrix raw files generated in this study have been deposited in the Gene Expression Omnibus database under accession code GSE196701.

## Immunoblot analysis
Protein lysates from human tubular cell line (HK-2) cells, kidneys and liver were subjected to SDS-PAGE and then to immunoblot analysis to detect the expression of IL-1 beta, HSP90, NGAL, HMGCR, LDLR, CD36, CPT1α, ATGL, p-PI3K, PI3K, p-Akt (Thr308), p-Akt (Ser473), t-Akt, p-P70(S6K), P70(S6K), SREBP-1, FAS, ACC1, SCD1, Caspase 3 and PPARγ following the standard procedure. The antibodies are listed in "Reporting summary" and Supplementary Table 4.

## Cell culture and treatments
The HK-2 (presented by Prof. Weidong Wang, Zhongshan School of Medicine, Sun Yat-sen University) cells were cultured in DMEM/F12 medium supplemented with 10% fetal bovine serum and 1% penicillin/streptomycin (all from Life Technologies, Carlsbad, CA) and maintained at 37 °C in 5% $CO_2$ atmosphere. Specifically, HK-2 cells were subjected to starvation in a serum-free medium overnight, then treat with OA (O1383, Sigma-Aldrich) (120 μM) and insulin 2.4 μM for 24 h, and separately subjected to treated with MK-2206 2HCl (S1078, Selleck Chemicals) 1 and 10 μM, CCI-779 (NSC 683864, Selleck) 1 and 10 μM for 24 h. The control group always cultured in a serum-free medium. The experiment was repeated three times. HK-2 cells were stained using SREBP-1 (1:200), BODIPY (1:1000) antibodies and secondary antibodies conjugated with Alexa Fluor. These cells were counterstained with DAPI, and their fluorescent signals were visualized using fluorescence microscope (LSM 800, Zeiss).

## Statistical analysis
All analyses are expressed as means ± SEM and analyzed using the statistical package for the GraphPad Prism 9.2 (Inc., La Jolla, CA) and the statistical package IBM SPSS Statistics software (SPSS) (Versions 22.0) (Inc., Chicago, IL). Significance tests were two-tailed, One-way ANOVA followed by Tukey's test was performed. In all statistical comparisons, *p* value < 0.05 was used to indicate a statistically significant difference.

## Reporting summary
Further information on research design is available in the Nature Portfolio Reporting Summary linked to this article.

# Data availability
Source data contained the raw data underlying the following types of display items: any reported means/averages in bar charts, and tables, and uncropped versions of any gels or blots, labeled with the relevant panel and identifying information. The affymetrix raw files generated in this study have been deposited in the GEO database under accession code GSE196701. Source data are provided with this paper.

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

## Acknowledgements

W.C. is supported by National Key Research and Development Program of China (Grant No. 2019YFA0801403), National Nature Science Foundation of China (Grant Nos. 82170261, 81741117, 81970219), Guangdong Basic and Applied Basic Research Foundation (Grant Nos: 2021A1515011005, 2021A1515110233 and 2021B1212040006). X.L. is supported by National Nature Science Foundation of China (Grant No. 82000250) and China Postdoctoral Science Foundation (Grant No: 2020M672976). J.T. is supported by China Postdoctoral Science Foundation (Grant No.: 2021M693613). The funders had no role in study design, data collection and analysis, decision to publish, or preparation of the manuscript. We thank Prof. Weidong Wang (Zhongshan School of Medicine, Sun Yat-sen University) for the gift of HK-2 cell.

## Author contributions

T.H. and W.C. designed the study and wrote the paper; T.H., T.W., Y.W., X.L., J.T., C.S., S.X. and Z.F. performed the primary experiments and analyzed the data; S.G. and H.L. provided technical assistance; T.H. and T.W. wrote the manuscript; W.C. is the guarantor of this work and, as such, had full access to all the data in the study and takes responsibility for the integrity of the data and the accuracy of the data analysis. All authors read and approved the final manuscript.

## Competing interests

The authors declare no competing interests.
