## [Peer Review File · Nature Communications]

Long-term statins administration exacerbates diabetic nephropathy via ectopic fat deposition in diabetic miceREVIEWER COMMENTS

Reviewer #1 (Remarks to the Author):

The authors have provided extensive studies indicating that in mouse models of diabetes, long term treatment with either of two statins significantly exacerbated diabetic kidney disease, associated with increased lipid deposition in the kidney and activation of SREBP-1, which increased renal fatty acid synthesis. In general, the studies appear well performed and the results are consistent with the overall hypothesis.

1) There is a concern with some of the data, especially in figure 1, that there might be a survivor bias, given that the studies were apparently performed at the end of the 50 week study, at a time whether were < 25% of the original statin treated cohort surviving. The studies should be repeated at an earlier time when there was less discrepancy in the survival ratio between treated and untreated mice. In this regard, all of the figure legends should indicate the age at which the mice were studied.

2) Were there differences in glomerular basement thickening in figure 2C. Also, it would be helpful to quantitate the amount of podocyte effacement.

3) In figure 7, there is a highlight of SREBP-1 in glomeruli, but FAS appears to be largely confined to tubules. Please comment.

4) The in vitro studies in figure 8 indicate that insulin can activate mTOR and increase SREBP-1 and lipid. It would be important to show that the statins had the same effect, indicating a role for PPARgamma activation.

Reviewer #2 (Remarks to the Author):

General comments:

This paper by Huang et al, investigates the effect of statin treatment on diabetic nephropathy in mice.

The study include several mouse models and multiple measurements. It appears to be well-performed. Nevertheless, there are several queries the authors are asked to consider.

- In general, please include the number of n (animals/measurements) in all figures, tables and method sections.

- The IHC pictures and the TEM are very small and they are taken at low magnification. Thus, it is difficult to see the tubule structures and the labeling. Please take pictures at a higher magnification and show a bigger area of the sections. It is difficult for the reader to evaluate the structures and the stainings (for IHC).

- Please add scalebar and arrows to the pictures.

- Please in general describe the labeling and specify which tubules are labelled and whether the labeling is in accordance with known litterature. Does all antibodies show specific labeling?

- Please specify which part of the kidney (cortex, OM, IM) is shown on the respective pictures.

- The authors used DAB – why is the labeling red and not brown in many of the pictures?.

- The discussion is very long. Please shorten and do not refer to figures in this section. And refer to figure 9.

- There are many abbreviations which are not specified.

Specific comments:

L 125: How is statins administrated ? in the food?

L 120: Specify Db/m and db/db: Is the Db/m wildtype or heterozygote ?

Why have the authors not included a group where control animals are administrated statins?

Specify number of animals in each experiment.

L 132; L144: Specify “some kidney and liver tissues”. Number of n for kidney and liver samples. And number of animals for serum sampling.

L147: What is KK-Ay mice. Are the controls littermates? If not, do they have the same background?

L157: Specify HED group, number of n.

L164: administrated with statins once a day – is this by gavage? Have the authors considered the turnover of the respective statins and whether this would influence the results.

L198: specify number of days in metabolic cages.

L211: Specify in which mouse models FFA was measured.

L246: DAB has been used. Why are labeling in many of the pictures red and not brown?

L266: For DHE staining – specify which animals models and number of n.

L304: Are all data normal distributed?

Figure 1: Show in panel a number of days and when the animals were in metabolic cages.

Suppl figure 1: The Masson trichrome staining looks over stained. Please show pictures with less intensity.

The authors truly appreciate the reviewers' time and effort in reviewing our manuscript, and the supportive comments on our findings. In this submission, some changes were made based on the comments from editor and reviewer. Descriptions of these changes are underlined in the manuscript. Specific critiques raised by each reviewer are addressed below. With these revisions, we hope the reviewer would find that the revised manuscript is sufficiently improved for publication.

Reviewer #1:

The authors have provided extensive studies indicating that in mouse models of diabetes, long term treatment with either of two statins significantly exacerbated diabetic kidney disease, associated with increased lipid deposition in the kidney and activation of SREBP-1, which increased renal fatty acid synthesis. In general, the studies appear well performed and the results are consistent with the overall hypothesis.

RESPONSE: The authors thank the Reviewer for his/her overall supportive comments. We try our best to revise the manuscript according to your suggestions and answer every question from you.

1) There is a concern with some of the data, especially in figure 1, that there might be a survivor bias, given that the studies were apparently performed at the end of the 40weeks study, at a time whether were < 25% of the original statin treated cohort surviving. The studies should be repeated at an earlier time when there was less discrepancy in the survival ratio between treated and untreated mice. In this regard, all of the figure legends should indicate the age at which the mice were studied.

RESPONSE: Thank you for your valuable suggestion. We agree with you that studies of long-term drug administration may have survivor bias. Generally, we first observe and collect the data of the experimental endpoint in order to improve research efficiency when we study the long-term effects of drugs.

Although we did not collect mouse kidneys for analysis at intermediate time points, we collected mouse urine at each time point, enabling us to effectively observe renal injury in mice during the whole experiment. Therefore, we supplemented the level of Kidney injury molecule 1 (KIM-1, also known as TIM-1) in mouse urine at each time point to investigate the renal injury of mice. As shown in Fig. 2n, the level of mouse urinary KIM1 in statin-treated group began to increase from the 19th week, which was consistent with the data of ACR. The urinary KIM1 in statin-treated groups kept a constant rise and were significantly higher than that in Db group at the late stage of statin administration. Furthermore, we supplemented the data of mouse GFR and renal pathology in the short-term (10 weeks) statin administration experiments to evaluate early renal injury. As shown in Supplemental Figure 10, short-term statin treatment reduced the survival rate of mice (a) and resulted in certain degree of renal injury (b-d). Therefore, we believe that survivor bias may not be the main determinant factor for us to analyze and consider the aggravated renal injury caused by long-term statins treatment in *db/db* mice. It is worth mentioning that our findings in animal studies are consistent with the latest retrospective study published in *JAMA Intern Med*, which showed that statin use was associated with diabetes progression in patients based on the data of statins used in diabetes patients for 12 years¹. We also qualitatively measured urinary hemoglobin (Supplementary figure 8a) to exclude the hemolytic side effects caused by myopathy during statin treatment, indicating that the time of taking statins is reasonable.

In addition, we add the information of mouse age in figure legends according to your suggestions to make the data clearer.

2) Were there differences in glomerular basement thickening in figure 2C. Also, it would be helpful to quantitate the amount of podocyte effacement.

RESPONSE: As the reviewer suggested, the thickness of glomerular basement and the amount of podocyte are important indicators for glomerular function ².

The data of glomerular structure detected by transmission electron microscopy (TEM) qualitatively indicated that statin treatment led to glomerular basement membrane thickening as shown in Fig. 2c. To clarify this further, the qualitative data of PASM staining (Fig. 2d) was used to evaluate glomerular basement membrane. The above data showed that the glomerular basement membrane was significantly thickened in statin-treated groups.

Furthermore, WT1 immunofluorescence analysis was used to quantify the podocytes in glomerulus as suggested by the reviewer. The data in Fig. 2e and 2i illustrated that long-term statin treatment reduced the number of glomerular podocytes in diabetes mice. Nephritin forms an integral part of podocytes, which—together with endothelial cells and the basement—form the glomerular filtration barrier³. The quantitative data of Nephritin immunofluorescence analysis (Fig. 2f & 2j) showed that long-term statin treatment occurs early in glomerular injury in diabetes mice.

3) In figure 7, there is a highlight of SREBP-1 in glomeruli, but FAS appears to be largely confined to tubules. Please comment.

RESPONSE: Thank you for your valuable comments. The transcription factor SREBP-1 and its target genes FAS and SCD1 play important roles in diabetes nephropathy disease progression^{2,4}. It has been reported that SREBP-1-mediated fatty acid synthesis leads to lipid deposition⁵, which may be the reason why SREBP-1 accelerates the progression of diabetes nephropathy. Although SREBP-1 and its target genes are involved in diabetes nephropathy, they are lowly expressed in kidney under physiological conditions (Fig. 7a). However, SREBP-1 and its target gene FAS and SCD1 are highly expressed in glomeruli and renal tubules in diabetes. In order to more clearly show the expression of FAS and SCD1 in glomeruli, we replaced the relevant images with high-power glomerular images in the new manuscript (Fig. 7b, c). Here, we also added the data indicating the high expression of SREBP-1 and its target genes FAS and SCD1 in renal tubular epithelial cells of diabetes nephropathy,

as shown in the figure below.

4) The *in vitro* studies in figure 8 indicate that insulin can activate mTOR and increase SREBP-1 and lipid. It would be important to show that the statins had the same effect, indicating a role for PPAR gamma activation.

RESPONSE: Thank you for your important comment and advice. Our data indicated that insulin upregulating SREBP-1 expression to promote fatty acid synthesis may be dependent on the PI3K-AKT-mTOR signaling pathway (Fig. 8b & 8c). PPAR γ can promote fatty acid uptake, triglyceride formation and storage in lipid droplets, thereby increasing insulin sensitivity and glucose metabolism. As you suggested, it is necessary to explore whether statins have the same effect *in vitro*, indicating a role for PPAR gamma activation. Our data suggests that long-term statin administration can significantly activate SREBP-1 in *db/db* mice (Fig. 7a & 7d), as does insulin stimulation of HK2 cells *in vitro* experiments. Unfortunately, we stimulated HK2 cells with statin (Supplementary Fig. 9a & 9b), but did not increase SREBP-1 and PPAR gamma as significantly as when stimulated with insulin. We think that it is difficult to simulate the complex state of hyperglycemia, hyperlipidemia, hyperinflammation and hyperinsulinemia in diabetes *in vitro* and even more difficult to simulate prolonged drug stimulation in mice treated with long-term medication, thus making it difficult to obtain consistent *in vivo* results.

Reviewer #2:

General comments:

This paper by Huang et al, investigates the effect of statin treatment on diabetic nephropathy in mice. The study includes several mouse models and multiple measurements. It appears to be well-performed. Nevertheless, there are several queries the authors are asked to consider.

RESPONSE: The authors really appreciate the reviewers' effort in reviewing our manuscript and supportive comments to our findings. We try our best to address all questions and suggestions point by point.

In general, please include the number of n (animals/measurements) in all figures, tables and method sections.

RESPONSE: Thank you for your suggestion and sorry for the confusion if there is any. We have added the number of *n* (animals/measurements) in all figures, tables and method sections according to your advice.

The IHC pictures and the TEM are very small and they are taken at low magnification. Thus, it is difficult to see the tubule structures and the labeling. Please take pictures at a higher magnification and show a bigger area of the sections. It is difficult for the reader to evaluate the structures and the staining (for IHC).

RESPONSE: Thank you for these important comments. As suggested, we have replaced some images with higher magnification to clearly illustrate the structures and the staining in the new manuscript, including 400 X magnification IHC images of 4-HNE, NGAL, IL-1 beta. In addition, the 9700X magnification TEM images have been added in the new manuscript to show more lipid droplets. Some images were replaced and marked with arrows for more visual presentation (Fig. 5e).

The IHC staining of COL1 and SMA was mainly used to show the extent of fibrosis. The 400X magnification image may not be beneficial to show the fibrotic area. Therefore, the original 200X magnification images were retained.

Please add scalebar and arrows to the pictures.

RESPONSE: Thank you for your suggestion and sorry for the confusion if there is any. As your suggestion, we have supplemented the scalebar in Fig 4b, Fig 5g, Fig 8a, and added arrows in Fig 4b, Fig 5e to improve the accuracy and visual presentation of the data.

Please in general describe the labeling and specify which tubules are labelled and whether the labeling is in accordance with known literature.

Do all antibodies show specific labeling?

RESPONSE: Thank you for your professional advice. The specificity of antibody is the essential requirement to ensure reliability and accuracy of research data. All the antibodies were selected in our studied with reference to authoritative literature, and were purchased from regular companies. Our experimental data also reflect the specificity of the antibody with accurate subcellular localization of the target proteins. The key Information for all antibodies is described in the manuscript or the supplementary materials (Supplementary Table 4). If necessary, we can provide further more information about the antibodies to ensure the stability and reliability of experiments in our research.

Please specify which part of the kidney (cortex, OM, IM) is shown on the respective pictures.

RESPONSE: Thank you very much for your suggestion. The specific part of the kidney shown in all figures is the cortex. We have added the relevant descriptions to the figure legends in the new manuscript, such as Fig 1. g, Fig 2., Fig 3. e-g, Fig 4., Fig 5., Fig 6., Fig 7. And Fig 9.

The authors used DAB – why is the labeling red and not brown in many of the pictures?

RESPONSE: Thank you for your careful consideration and sorry for the

confusion if there is any. DAB (3,3'-diaminobenzidine) is oxidized in the presence of peroxidase and hydrogen peroxide resulting in the deposition of a brown, alcohol-insoluble precipitate at the site of enzymatic activity. DAB produces a dark brown reaction product that can be used in immunohistochemistry. Some IHC images in my data were dark red, possibly due to the contrast settings of the microscope at the time of the photo, which resulted in high contrast in some of the images. Therefore, we rescaled the contrast of some of the pictures marked red, such as Figures 6b and 6d, to show the pictures labeling brown.

The discussion is very long. Please shorten and do not refer to figures in this section. And refer to figure 9.

RESPONSE: We thank the reviewer for his/her careful consideration. We have revised the Discussion section according to the Reviewer's suggestion. The number of words in the discussion part is reduced from 2132 to 1578, which meets the writing requirements of the journal. In addition, only the figure 10 (Figure 9 in the old manuscript) has been referred to in the Discussion section of the new manuscript.

There are many abbreviations which are not specified.

RESPONSE: Thank you for your suggestion and sorry for the confusion if there is any. All abbreviations in the original text have been specified and revised with full information, and all changes in the new manuscript are shown with tracked mark.

Specific comments:

L 125: How is statins administrated? in the food?

RESPONSE: Sorry for the confusion, we should describe the experimental method more clearly. Statins were dissolved or suspended in 0.9% saline and administered by gavage once a day. In the new manuscript, we have added a

description of drug administration in the section of Animal model and treatment.

L 120: Specify Db/m and db/db: Is the Db/m wildtype or heterozygote?

Why have the authors not included a group where control animals are administered statins? Specify number of animals in each experiment.

RESPONSE: The *db/db* mice serve as a good model for type 2 diabetes, characterized by hyperinsulinemia and progressive hyperglycemia. The *db/m* is the heterozygous mouse without diabetes and is often used as the control for *db/db* mice. We are sorry for the confusion if there is any, and have added some information about *db/m* mice in “Animal model and treatment” of the new manuscript.

The *db/m* mice were used as the control group administered statins in our research. The survival curve (Fig.1C) and PAS staining and Masson staining (Supplementary Fig. 1) showed that long-term statin administration had no effect on the survival and the renal structures of *db/m* mice.

The number of animals in each experiment was added to the figure legends in the new manuscript.

L 132; L144: Specify “some kidney and liver tissues”. Number of n for kidney and liver samples. And number of animals for serum sampling.

RESPONSE: Sorry for the confusion. As suggested by reviewer, the number of animals for kidney, liver and serum sampling was added to the figure legends in the new manuscript.

L147: What is KK-Ay mice. Are the controls littermates? If not, do they have the same background?

RESPONSE: The KK mouse is a polygenic mouse model of T2DM. KK-Ay mice are a cross between diabetic KK and lethal yellow (Ay) mice, and carry a heterozygous mutation of the agouti gene⁶. The severity of hyperglycemia and insulin resistance is exacerbated by the introduction of Ay allele into the KK

background. KK-Ay mice exhibit obesity and hyperglycemia, as well as albuminuria. Renal histological changes, such as podocyte loss, diffuse mesangial expansion with mesangial cell proliferation, and segmental sclerosis in KK-Ay mice⁷, are more severe than those that develop in KK mice. The genetic background of KK mice and KK-Ay mice is the inbred mouse strain of C57BL/6J, which is always used as the control for KK mice or KK-Ay mice.

L157: Specify HFD group, number of n.

RESPONSE: Sorry for the confusion. As suggested by reviewer, the animal number of HFD group has been added in the new manuscript, such as L163, L168.

L164: administrated with statins once a day – is this by gavage? Have the authors considered the turnover of the respective statins and whether this would influence the results.

RESPONSE: Thank you for raising this important question. Statins were dissolved or suspended in 0.9% saline and administered by gavage once a day. In the new manuscript, we have added a description of drug administration in the section of Animal model and treatment.

We did consider the effects of statin metabolic turnover on drug action and efficacy. These statins are commonly used clinically and have robust lipid-lowering effects. The metabolic turnover between different statins is indeed different, which is manifested by inconsistent effective doses of different statins. We verified the required dose of two statins in mice according to the literature and found that the lipid-lowering effect of 10 mg/kg Ato was comparable to that of 20 mg/kg Rosu in both HFD models. In addition, we observed lipid-lowering effects of different doses of statins in several diabetic mouse models, suggesting that the metabolic turnover of individual statins would do not influence drug efficacy and experimental results.

L198: specify number of days in metabolic cages.

RESPONSE: Thank you for your suggestion. The number of days in metabolic cages has been added in the new manuscript, L194. To assess urinary albumin excretion, mice urine samples were collected by metabolic cage in 12 hours per four weeks and stored at -80°C. Mice only need to be placed in the metabolic cage for 12 hours each time, and the urine collected for 12 hours can meet the determination of ACR.

L211: Specify in which mouse models FFA was measured.

RESPONSE: Sorry for the confusion. We used the *db/db* mouse model for measuring the tissues FFAs uptake by fluorescence imaging. As suggested, we have specified the mouse model in the section of “Tissues FFAs uptake fluorescence imaging” in the new manuscript, L211.

L246: DAB has been used. Why are labeling in many of the pictures red and not brown?

RESPONSE: Thank you for your careful consideration and sorry for the confusion if there is any. DAB (3,3'-diaminobenzidine) is oxidized in the presence of peroxidase and hydrogen peroxide resulting in the deposition of a brown, alcohol-insoluble precipitate at the site of enzymatic activity. DAB produces a dark brown reaction product that can be used in immunohistochemistry. Some IHC images in my data were dark red, possibly due to the contrast settings of the microscope at the time of the photo, which resulted in high contrast in some of the images. Therefore, we rescaled the contrast of some of the pictures marked red, such as Figures 6b and 6d, to show the pictures labeling brown.

L266: For DHE staining – specify which animals models and number of n.

RESPONSE: Sorry for the confusion. We used the *db/db* mouse model for DHE

staining, and the animal number of n was 6. As suggested, we have specified the mouse model and add the information of animal number in the section of “Dihydroethidium (DHE) staining” in the new manuscript, L274.

L304: Are all data normal distributed?

RESPONSE: The normal distribution of data is a prerequisite for many analysis methods. Before performing analysis operations such as variance analysis, independent sample t-test, regression analysis, etc., the normality of the data must be analyzed to ensure that the method is selected correctly. The variables observed in our studies are normally distributed.

Figure 1: Show in panel a number of days and when the animals were in metabolic cages.

RESPONSE: Thank you for your suggestion. The number of days in metabolic cages has been added in the new manuscript, L195. To assess urinary albumin excretion, mice urine samples were collected by metabolic cage in 12 hours per four weeks and stored at -80°C. Mice only need to be placed in the metabolic cage for 12 hours each time, and the urine collected for 12 hours can meet the determination of ACR.

As suggested by the reviewer, we have added this information about the days and the time to the panel in Fig. 1a of the new manuscript.

Suppl figure 1: The Masson trichrome staining looks over stained. Please show pictures with less intensity.

RESPONSE: Thank you for your suggestion. We have replaced the Masson's trichrome stained images with lower intensity images, please refer to Supplementary Figure 1 in the new manuscript.

In all, I found the reviewer's comments to be quite helpful, and I revised the paper point-by-point. We earnestly appreciate the Reviewers' work and hope

that the corrections will meet with approval.

References

- 1 Mansi, I. A. *et al.* Association of Statin Therapy Initiation With Diabetes Progression: A Retrospective Matched-Cohort Study. *JAMA Intern Med* **181**, 1562-1574, doi:10.1001/jamainternmed.2021.5714 (2021).
- 2 Fu, Y. *et al.* Elevation of JAML Promotes Diabetic Kidney Disease by Modulating Podocyte Lipid Metabolism. *Cell Metab* **32**, 1052-1062 e1058, doi:10.1016/j.cmet.2020.10.019 (2020).
- 3 Kandasamy, Y., Smith, R., Lumbers, E. R. & Rudd, D. Nephricin - a biomarker of early glomerular injury. *Biomark Res* **2**, 21, doi:10.1186/2050-7771-2-21 (2014).
- 4 Mitrofanova, A., Burke, G., Merscher, S. & Fornoni, A. New insights into renal lipid dysmetabolism in diabetic kidney disease. *World J Diabetes* **12**, 524-540, doi:10.4239/wjd.v12.i5.524 (2021).
- 5 Shimano, H. & Sato, R. SREBP-regulated lipid metabolism: convergent physiology - divergent pathophysiology. *Nat Rev Endocrinol* **13**, 710-730, doi:10.1038/nrendo.2017.91 (2017).
- 6 Suto, J., Matsuura, S., Imamura, K., Yamanaka, H. & Sekikawa, K. Genetic analysis of non-insulin-dependent diabetes mellitus in KK and KK-Ay mice. *Eur J Endocrinol* **139**, 654-661, doi:10.1530/eje.0.1390654 (1998).
- 7 Kitada, M., Ogura, Y. & Koya, D. Rodent models of diabetic nephropathy: their utility and limitations. *Int J Nephrol Renovasc Dis* **9**, 279-290, doi:10.2147/IJNRD.S103784 (2016).

REVIEWER COMMENTS

Reviewer #1 (Remarks to the Author):

The authors have been very responsive to the previous reviews and have clarified points raised with further results. I have no further comments and congratulate the authors on a very interesting and timely study.

Reviewer #2 (Remarks to the Author):

In general, please include the number of n (animals/measurements) in all figures, tables and method sections.

RESPONSE: Thank you for your suggestion and sorry for the confusion if there is any. We have added the number of n (animals/measurements) in all figures, tables and method sections according to your advice.

N was not included in all method sections and tables?

What is meant by biologically independent mice?

The IHC pictures and the TEM are very small and they are taken at low magnification. Thus, it is difficult to see the tubule structures and the labeling. Please take pictures at a higher magnification and show a bigger area of the sections. It is difficult for the reader to evaluate the structures and the staining (for IHC).

RESPONSE: Thank you for these important comments. As suggested, we have replaced some images with higher magnification to clearly illustrate the structures and the staining in the new manuscript, including 400 X magnification IHC images of 4-HNE, NGAL, IL-1 beta. In addition, the 9700X magnification TEM images have been added in the new manuscript to show more lipid droplets. Some images were replaced and marked with arrows for more visual presentation (Fig. 5e).

The IHC staining of COL1 and SMA was mainly used to show the extent of fibrosis. The 400X magnification image may not be beneficial to show the fibrotic area. Therefore, the original 200X magnification images were retained.

Although a higher magnification has been shown for a few pictures, it is still difficult to evaluate the staining/labeling on many of the pictures. The authors did not show a larger area of the sections. Arrows should be added and the labeling/staining should be described (which tubules, subcellular localization, which areas in the ECM) with references to known literature for the used antibodies.

Please add scalebar and arrows to the pictures.

RESPONSE: Thank you for your suggestion and sorry for the confusion if there is any. As your suggestion, we have supplemented the scalebar in Fig 4b, Fig 5g, Fig 8a, and added arrows in Fig 4b, Fig 5e to improve the accuracy and visual presentation of the data.

Arrows should be added to all pictures where relevant. Otherwise difficult to follow for the reader.

Please in general describe the labeling and specify which tubules are labelled and whether the labeling is in accordance with known literature. Do all antibodies show specific labeling?

RESPONSE: Thank you for your professional advice. The specificity of antibody is the essential requirement to ensure reliability and accuracy of research data. All the antibodies were selected in our studied with reference to authoritative literature, and were purchased from regular companies. Our experimental data also reflect the specificity of the antibody with accurate subcellular localization of the target proteins. The key Information for all antibodies is described in the manuscript or the supplementary materials (Supplementary Table 4). If necessary, we can provide further more information about the antibodies to ensure the stability and reliability of experiments in our research.

You need to describe the labeling (ab) or staining – which tubules are labelled and the subcellular localizations, which areas of the ECM. This is extremely important. Please include references for the antibodies in the text.

Please specify which part of the kidney (cortex, OM, IM) is shown on the respective pictures.

RESPONSE: Thank you very much for your suggestion. The specific part of the kidney shown in all figures is the cortex. We have added the relevant descriptions to the figure legends in the new manuscript, such as Fig 1. g, Fig 2., Fig 3. e-g, Fig 4., Fig 5., Fig 6., Fig 7. And Fig 9.

Ok

The authors used DAB – why is the labeling red and not brown in many of the pictures?

RESPONSE: Thank you for your careful consideration and sorry for the confusion if there is any. DAB (3,3'-diaminobenzidine) is oxidized in the presence of peroxidase and hydrogen peroxide resulting in the deposition of a brown, alcohol-insoluble precipitate at the site of enzymatic activity. DAB produces a dark brown reaction product that can be used in immunohistochemistry. Some IHC images in my data were dark red, possibly due to the contrast settings of the microscope at the time of the photo, which resulted in high contrast in some of the images. Therefore, we rescaled the contrast of some of the pictures marked red, such as Figures 6b and 6d, to show the pictures labeling brown.

Ok. Although still red. Were all pictures taken with the same settings?

The discussion is very long. Please shorten and do not refer to figures in this section. And refer to figure 9.

RESPONSE: We thank the reviewer for his/her careful consideration. We have revised the Discussion section according to the Reviewer's suggestion. The number of words in the discussion part is reduced from 2132 to 1578, which meets the writing requirements of the journal. In addition, only the figure 10 (Figure 9 in the old manuscript) has been referred to in the Discussion section of the new manuscript.

Ok.

There are many abbreviations which are not specified.

RESPONSE: Thank you for your suggestion and sorry for the confusion if there is any. All abbreviations in the original text have been specified and revised with full information, and all changes in the new manuscript are shown with tracked mark.

Ok

Specific comments:

L 125: How is statins administrated? in the food?

RESPONSE: Sorry for the confusion, we should describe the experimental method more clearly. Statins were dissolved or suspended in 0.9% saline and administered by gavage once a day. In the new manuscript, we have added a description of drug administration in the section of Animal model and treatment.

Ok.

L 120: Specify Db/m and db/db: Is the Db/m wildtype or heterozygote?

Why have the authors not included a group where control animals are administered statins? Specify number of animals in each experiment.

RESPONSE: The db/db mice serve as a good model for type 2 diabetes, characterized by hyperinsulinemia and progressive hyperglycemia. The db/m is the heterozygous mouse without diabetes and is often used as the control for db/db mice. We are sorry for the confusion if there is any, and have added some information about db/m mice in “Animal model and treatment” of the new manuscript.

The db/m mice were used as the control group administered statins in our research. The survival curve (Fig.1C) and PAS staining and Masson staining (Supplementary Fig. 1) showed that long-term statin administration had no effect on the survival and the renal structures of db/m mice.

The number of animals in each experiment was added to the figure legends in the new manuscript.

Ok.

L 132; L144: Specify “some kidney and liver tissues”. Number of n for kidney and liver samples. And number of animals for serum sampling.

RESPONSE: Sorry for the confusion. As suggested by reviewer, the number of animals for kidney, liver and serum sampling was added to the figure legends in the new manuscript.

Ok

L147: What is KK-Ay mice. Are the controls littermates? If not, do they have the same background?

RESPONSE: The KK mouse is a polygenic mouse model of T2DM. KK-Ay mice are a cross between diabetic KK and lethal yellow (Ay) mice, and carry a heterozygous mutation of the agouti gene⁶. The severity of hyperglycemia and insulin resistance is exacerbated by the introduction of Ay allele into the KK background. KK-Ay mice exhibit obesity and hyperglycemia, as well as albuminuria. Renal histological changes, such as podocyte loss, diffuse mesangial expansion with mesangial cell proliferation, and segmental sclerosis in KK-Ay mice⁷, are more severe than those that develop in KK mice. The genetic background of KK mice and KK-Ay mice is the inbred mouse strain of C57BL/6J, which is always used as the control for KK mice or KK-Ay mice.

Please include some of this description in the paper to help the reader.

L157: Specify HFD group, number of n.

RESPONSE: Sorry for the confusion. As suggested by reviewer, the animal number of HFD group has been added in the new manuscript, such as L163, L168.

Please specify HFD group.

L164: administrated with statins once a day – is this by gavage? Have the authors considered the turnover of the respective statins and whether this would influence the results.

RESPONSE: Thank you for raising this important question. Statins were dissolved or suspended in 0.9% saline and administered by gavage once a day. In the new manuscript, we have added a description of drug administration in the section of Animal model and treatment.

We did consider the effects of statin metabolic turnover on drug action and efficacy. These statins are commonly used clinically and have robust lipid-lowering effects. The metabolic turnover between different statins is indeed different, which is manifested by inconsistent effective doses of different statins. We verified the required dose of two statins in mice according to the literature and found that the lipid-lowering effect of 10 mg/kg Ato was comparable to that of 20 mg/kg Rosu in both HFD models. In addition, we observed lipid-lowering effects of different doses of statins in several diabetic mouse models, suggesting that the metabolic turnover of individual statins would do not influence drug efficacy and experimental results.

Ok.

L198: specify number of days in metabolic cages.

RESPONSE: Thank you for your suggestion. The number of days in metabolic cages has been added in the new manuscript, L194. To assess urinary albumin excretion, mice urine samples were collected by metabolic cage in 12 hours per four weeks and stored at -80° C. Mice only need to be placed in the metabolic cage for 12 hours each time, and the urine collected for 12 hours can meet the determination of ACR.

Ok.

L211: Specify in which mouse models FFA was measured.

RESPONSE: Sorry for the confusion. We used the db/db mouse model for measuring the tissues FFAs uptake by fluorescence imaging. As suggested, we have specified the mouse model in the section of “Tissues FFAs uptake fluorescence imaging” in the new manuscript, L211.

Ok.

L266: For DHE staining – specify which animals models and number of n.

RESPONSE: Sorry for the confusion. We used the db/db mouse model for DHE staining, and the animal number of n was 6. As suggested, we have specified the mouse model and add the information of animal number in the section of “Dihydroethidium (DHE) staining” in the new manuscript, L274.

Ok.

L304: Are all data normal distributed?

RESPONSE: The normal distribution of data is a prerequisite for many analysis methods. Before performing analysis operations such as variance analysis, independent sample t-test, regression analysis, etc., the normality of the data must be analyzed to ensure that the method is selected correctly. The variables observed in our studies are normally distributed.

Ok

Figure 1: Show in panel a number of days and when the animals were in metabolic cages.

RESPONSE: Thank you for your suggestion. The number of days in metabolic cages has been added in the new manuscript, L195. To assess urinary albumin excretion, mice urine samples were collected by metabolic cage in 12 hours per four weeks and stored at -80° C. Mice only need to be placed in the metabolic cage for 12 hours each time, and the urine collected for 12 hours can meet the determination of ACR.

As suggested by the reviewer, we have added this information about the days and the time to the panel in Fig. 1a of the new manuscript.

Ok

Suppl figure 1: The Masson trichrome staining looks over stained. Please show pictures with less intensity.

RESPONSE: Thank you for your suggestion. We have replaced the Masson's trichrome stained images with lower intensity images, please refer to Supplementary Figure 1 in the new manuscript.

Ok

The authors truly appreciate the reviewers' time and effort in reviewing our manuscript, and the supportive comments on our findings. In this submission, some changes were made based on the comments from editor and reviewers. Descriptions of these changes are underlined in the manuscript. Response to the comments is addressed below:

Reviewer 1:

The authors have been very responsive to the previous reviews and have clarified points raised with further results. I have no further comments and congratulate the authors on a very interesting and timely study.

RESPONSE: Thank you for your kindness in reviewing our manuscript and supportive comments to our work.

Reviewer 2:

(1) In general, please include the number of n (animals/measurements) in all figures, tables and method sections.

First-round RESPONSE: Thank you for your suggestion and sorry for the confusion if there is any. We have added the number of n (animals/measurements) in all figures, tables and method sections according to your advice.

N was not included in all method sections and tables?

What is meant by biologically independent mice?

Second-round RESPONSE: Thank you for your suggestion and sorry for the confusion. We have re-added the number of n (animals/measurements) in all figures, tables and method sections according to your advice, such as L128, L131, L134, L190, L211, L220, etc.

The word "biologically independent mice" has all been replaced with "in each group".

(2) The IHC pictures and the TEM are very small and they are taken at low magnification. Thus, it is difficult to see the tubule structures and the labeling. Please take pictures at a higher magnification and show a bigger area of the sections. It is difficult for the reader to evaluate the structures and the staining (for IHC).

First-round RESPONSE: Thank you for these important comments. As suggested, we have replaced some images with higher magnification to clearly illustrate the structures and the staining in the new manuscript, including 400 X magnification IHC images of 4-HNE, NGAL, IL-1 beta. In addition, the 9700X magnification TEM images have been added in the new manuscript to show more lipid droplets. Some images were replaced and marked with arrows for more visual presentation (Fig. 5e).

The IHC staining of COL1 and SMA was mainly used to show the extent of fibrosis. The 400X magnification image may not be beneficial to show the fibrotic area. Therefore, the original 200X magnification images were retained.

Although a higher magnification has been shown for a few pictures, it is still difficult to evaluate the staining/labeling on many of the pictures. The authors did not show a larger area of the sections. Arrows should be and the labeling/staining should be described (which tubules, subcellular localization, which areas in the ECM) with references to known literature for the used antibodies.

Second-round RESPONSE: Thank you for your suggestions. The higher magnification images of **all** IHC data have been added to the newly revised manuscript, please review the updated figures. The IHC staining of COL1 and α SMA has all been replaced with high magnification images (Fig 3c and 3d). And, as you suggested, the positive expression information about subcellular localization and tubules in the **all** IHC staining data has been described in the figure legends. Moreover, the arrows have added to all pictures according to your suggestions.

(3) Please add scalebar and arrows to the pictures.

First-round RESPONSE: Thank you for your suggestion and sorry for the confusion if there is any. As your suggestion, we have supplemented the scalebar in Fig 4b, Fig 5g, Fig 8a, and added arrows in Fig 4b, Fig 5e to improve the accuracy and visual presentation of the data.

Arrows should be added to all pictures where relevant. Otherwise difficult to follow for the reader.

Second-round RESPONSE: Thank you for your suggestion. Arrows have added to **all** pictures to improve the accuracy and visual presentation of the data.

(4) Please in general describe the labeling and specify which tubules are labelled and whether the labeling is in accordance with known literature. Do all antibodies show specific labeling?

First-round RESPONSE: Thank you for your professional advice. The specificity of antibody is the essential requirement to ensure reliability and accuracy of research data. All the antibodies were selected in our studied with reference to authoritative literature, and were purchased from regular companies. Our experimental data also reflect the specificity of the antibody with accurate subcellular localization of the target proteins. The key Information for all antibodies is described in the manuscript or the supplementary materials (Supplementary Table 4). If necessary, we can provide further more information about the antibodies to ensure the stability and reliability of experiments in our research.

You need to describe the labeling (ab) or staining – which tubules are labelled and the subcellular localizations, which areas of the ECM. This is extremely important. Please include references for the antibodies in the

text.

Second-round RESPONSE: Thank you for your important suggestions. We have supplemented the labeling (ab) or staining information in the text of the new manuscript. In addition, we have summarized a table with a list of IHC antibodies, including which tubules are labelled and the subcellular localizations, references. Please review the **Supplementary Table 5** for details.

(5) The authors used DAB – why is the labeling red and not brown in many of the pictures?

First-round RESPONSE: Thank you for your careful consideration and sorry for the confusion if there is any. DAB (3,3'-diaminobenzidine) is oxidized in the presence of peroxidase and hydrogen peroxide resulting in the deposition of a brown, alcohol-insoluble precipitate at the site of enzymatic activity. DAB produces a dark brown reaction product that can be used in immunohistochemistry. Some IHC images in my data were dark red, possibly due to the contrast settings of the microscope at the time of the photo, which resulted in high contrast in some of the images. Therefore, we rescaled the contrast of some of the pictures marked red, such as Figures 6b and 6d, to show the pictures labeling brown.

Ok. Although still red. Were all pictures taken with the same settings?

Second-round RESPONSE: We used the same settings to photograph the IHC staining of each antibody, but the staining effect of each antibody was slightly different, resulting in some differences in the color of the IHC images. We reprocessed the two IHC images of **Fig 6 b** and **Fig 6d** to make the labeling brown.

(6) L147: What is KK-Ay mice. Are the controls littermates? If not, do they have the same background?

First-round RESPONSE: The KK mouse is a polygenic mouse model of T2DM.

KK-Ay mice are a cross between diabetic KK and lethal yellow (Ay) mice, and carry a heterozygous mutation of the agouti gene⁶. The severity of hyperglycemia and insulin resistance is exacerbated by the introduction of Ay allele into the KK background. KK-Ay mice exhibit obesity and hyperglycemia, as well as albuminuria. Renal histological changes, such as podocyte loss, diffuse mesangial expansion with mesangial cell proliferation, and segmental sclerosis in KK-Ay mice, are more severe than those that develop in KK mice. The genetic background of KK mice and KK-Ay mice is the inbred mouse strain of C57BL/6J, which is always used as the control for KK mice or KK-Ay mice.

Please include some of this description in the paper to help the reader.

Second-round RESPONSE: Sorry for the confusion. I have added some of this description in the text, please review the L155-L162 in the revised manuscript.

(7) L157: Specify HFD group, number of n.

First-round RESPONSE: Sorry for the confusion. As suggested by reviewer, the animal number of HFD group has been added in the new manuscript, such as L163, L168.

Please specify HFD group.

Second-round RESPONSE: The HFD model group were assigned to 6 groups: CON group, HFD group, HFD+Ato3 group (atorvastatin 3 mg/kg BW/day), HFD+Ato5 group (atorvastatin 5 mg/kg BW/day), HFD+Ato10 group (atorvastatin10 mg/kg BW/day) and HFD+Rosu20 group (Rosuvastatin 20 mg/kg BW/day). *n* =4 in each HFD model group (L174-L178).

In all, I found the reviewer's comments to be quite helpful, and I have revised the paper point-by-point. We appreciate the Reviewers' work and hope that the corrections may meet with approval.

REVIEWERS' COMMENTS

Reviewer #2 (Remarks to the Author):

(1)

N was not included in all method sections and tables?

What is meant by biologically independent mice?

Second-round RESPONSE: Thank you for your suggestion and sorry for the confusion. We have re-added the number of n (animals/measurements) in all figures, tables and method sections according to your advice, such as L128, L131, L134, L190, L211, L220, etc.

The word "biologically independent mice" has all been replaced with "in each group".

Ok

(2)

Although a higher magnification has been shown for a few pictures, it is still difficult to evaluate the staining/labeling on many of the pictures. The authors did not show a larger area of the sections. Arrows should be and the labeling/staining should be described (which tubules, subcellular localization, which areas in the ECM) with references to known literature for the used antibodies.

Second-round RESPONSE: Thank you for your suggestions. The higher magnification images of all IHC data have been added to the newly revised manuscript, please review the updated figures. The IHC staining of COL1 and α SMA has all been replaced with high magnification images (Fig 3c and 3d). And, as you suggested, the positive expression information about subcellular localization and tubules in the all IHC staining data has been described in the figure legends. Moreover, the arrows have added to all pictures according to your suggestions.

Specific comments for figures:

Figure 1: Please be aware that distal tubules do not have a brush border. It is not possible to identify a cell membrane on LM – only plasma membrane domains.

Figure 2: Panel c: please explain enlargement pictures and stars.

Panel d: PASM: Please relate the described structures (from line 404) to the arrows on the pictures. What do the arrows point at.

Panel f: Nephtrin. The arrows point at areas with no labeling?

Figure 3: Panel a: please describe the staining.

Panel b: describe what the arrows point at.

Panel d: the authors write that labeling is in fibroblasts. How do you know that it is fibroblasts? Arrows are also in the glomerulus?

Figure 4: Panel a: It is not possible to see the cell membrane?

Panel b: There are many green nuclei. In which cells are they located? On picture 4: What is the larger green area – artefact?

Panel c and d: it is not possible to identify vesicles.

Figure 5: Panel d: refer to the arrows. What do they point at?

Panel f: It is very difficult to see the stained structures/cells in the insets. Looks overstained.

Panel g: describe the arrows. What do they point at.

Figure 6: panel b-d. What are podocytes on the pictures? Some tubules are collapsed eg panel c picture 4: impossible to see the staining on this picture.

Figure 7: What are podocytes in the pictures. Be more specific what the arrows point at.

Figure 8: The authors write that SREBP-1 is located in the nucleus. This is not seen on the pictures?

Figure 9: panel c and d: Please describe the difference (if any) between the pictures (eg WT, STZ and STZ+ATO10).

Arrows should be added to all pictures where relevant. Otherwise difficult to follow for the reader.

Second-round RESPONSE: Thank you for your suggestion. Arrows have added to all pictures to improve the accuracy and visual presentation of the data.

Please also describe all arrows. What do they point at?

(4)

You need to describe the labeling (ab) or staining – which tubules are labelled and the subcellular localizations, which areas of the ECM. This is extremely important. Please include references for the antibodies in the text.

Second-round RESPONSE: Thank you for your important suggestions. We have supplemented the labeling (ab) or staining information in the text of the new manuscript. In addition, we have summarized a table with a list of IHC antibodies, including which tubules are labelled and the subcellular localizations, references. Please review the Supplementary Table 5 for details.

For supl table 5. I would include the references and the labeling description in the text and not in a table. Please be aware that it is not possible to identify lysosomes, golgi, ER etc on light microscopy.

(5)

Ok. Although still red. Were all pictures taken with the same settings?

Second-round RESPONSE: We used the same settings to photograph the IHC staining of each antibody, but the staining effect of each antibody was slightly different, resulting in some differences in the color of the IHC images. We reprocessed the two IHC images of Fig 6 b and Fig 6d to make the labeling brown.

Ok

(6)

Please include some of this description in the paper to help the reader.

Second-round RESPONSE: Sorry for the confusion. I have added some of this description in the text, please review the L155-L162 in the revised manuscript.

Ok

(7) L157: Specify HFD group, number of n.

Please specify HFD group.

Second-round RESPONSE: The HFD model group were assigned to 6 groups: CON group, HFD group, HFD+Ato3 group (atorvastatin 3 mg/kg BW/day), HFD+Ato5 group (atorvastatin 5 mg/kg BW/day), HFD+Ato10 group (atorvastatin10 mg/kg BW/day) and HFD+Rosu20 group (Rosuvastatin 20 mg/kg BW/day). n =4 in each HFD model group (L174-L178).

What is HFD the abbreviation for?

8) The authors write line 235 “The positive area was quantified with Image J...” Please write in details how the quantification of labeling has been performed. What are the criteria for selection of labeled tubules, criteria for background identification, are the same kind of tubules e.g. proximal tubules selected on the respective sections.

**Response to the comments of reviewers on
NCOMMS-21-49680B**

We truly appreciate the editors and reviewers for their time and effort in reviewing our manuscript. Your comments provided valuable suggestions for further revision and improvement of our manuscript. In this submission, some revisions were made based on the comments from editor and reviewers. Descriptions of these changes are underlined in the manuscript. Response to the comments is addressed below:

Reviewer #2:

(1) Although a higher magnification has been shown for a few pictures, it is still difficult to evaluate the staining/labeling on many of the pictures. The authors did not show a larger area of the sections. Arrows should be and the labeling/staining should be described (which tubules, subcellular localization, which areas in the ECM) with references to known literature for the used antibodies.

Second-round RESPONSE: Thank you for your suggestions. The higher magnification images of all IHC data have been added to the newly revised manuscript, please review the updated figures. The IHC staining of COL1 and α SMA has all been replaced with high magnification images (Fig 3c and 3d). And, as you suggested, the positive expression information about subcellular localization and tubules in the all IHC staining data has been described in the figure legends. Moreover, the arrows have added to all pictures according to your suggestions.

Specific comments for figures:

Figure 1: Please be aware that distal tubules do not have a brush border.

It is not possible to identify a cell membrane on LM – only plasma membrane domains.

Figure 2: Panel c: please explain enlargement pictures and stars.

Panel d: PASM: Please relate the described structures (from line 404) to the arrows on the pictures. What do the arrows point at.

Panel f: Nephryn. The arrows point at areas with no labeling?

Figure 3: Panel a: please describe the staining.

Panel b: describe what the arrows point at.

Panel d: the authors write that labeling is in fibroblasts. How do you know that it is fibroblasts? Arrows are also in the glomerulus?

Figure 4: Panel a: It is not possible to see the cell membrane?

Panel b: There are many green nuclei. In which cells are they located? On picture 4: What is the larger green area – artefact?

Panel c and d: it is not possible to identify vesicles.

Figure 5: Panel d: refer to the arrows. What do they point at?

Panel f: It is very difficult to see the stained structures/cells in the insets. Looks overstained.

Panel g: describe the arrows. What do they point at.

Figure 6: panel b-d. What are podocytes on the pictures? Some tubules are collapsed eg panel c picture 4: impossible to see the staining on this picture.

Figure 7: What are podocytes in the pictures. Be more specific what the arrows point at.

Figure 8: The authors write that SREBP-1 is located in the nucleus. This is not seen on the pictures?

Figure 9: panel c and d: Please describe the difference (if any) between the pictures (eg WT, STZ and STZ+ATO10).

Third-round RESPONSE: Thank you for your suggestion and sorry for the confusion if there is any. We are very pleased to answer the comments of the

Reviewer 2 one by one.

Figure 1: We agree with the reviewer's comment that distal tubules do not have a brush border. Therefore, the words "plasma membrane" in Line 935 has been replaced by "plasma membrane domains" in the revised manuscript as suggested.

Figure 2:

Panel c: The ultrastructure of renal cortex under transmission electron microscope was shown in the enlarged pictures, and the stars in pictures were used to marked the basement membrane thickening and foot process fusion.

We have made the corresponding revision in Line 949 of the revised manuscript.

Panel d: Panel d: We have related the described structures to the arrows in the Figure 2 panel d as shown from Line 178 to Line 182 in the new manuscript. The arrows point at the thickened basement membrane in the statin administration groups, which has been explained in Line 954-Line 956.

Panel f: We feel sorry for the confusion of the arrow marks in Figure 2 panel f. We have readjusted the position of the arrows to make a more accurate point at Nephrin positive staining in Figure 2 panel f.

Figure 3:

Panel a: As suggested by the reviewer, the description of Figure 3 panel a has been supplemented in Line 977 to Line 980 of the new manuscript.

Panel b: The arrows in Figure 3 panel b point at the fibrosis in kidney fibrotic interstitial lesions, which has been explained in Line 982-Line983.

Panel d: Panel d: Thank you for your important comments. α -Smooth muscle actin (α -SMA) is used as a marker for a subset of activated fibrogenic cells, myofibroblasts, which are regarded as important effector cells of renal fibrogenesis (Nat Rev Nephrol. 2019 Mar;15(3):144-158). As currently known, α -SMA is mainly expressed in vascular smooth muscle cells and myofibroblasts. Therefore, we agree with you that it is inaccurate to directly identify the labeled

cells in Figure 3 panel d as fibroblasts. It may be better to describe the fibrosis condition in Figure 3 panel d rather than to determine the cell type. We have made the corresponding revision in the new manuscript.

Figure 4:

Panel a: We agree with you that the cell membrane structure in the immunohistochemical section cannot be identified under the light microscope, and only the cellular region can be distinguished. In Figure 4 panel a, immunohistochemical staining of CD68 is used to identify and analyze the number of macrophages, and the membrane localization of CD68 in macrophage is not confirmed. We have made the corresponding modifications in the new manuscript.

Panel b: The green nuclei were mainly located at the inflammatory cells in the tubulointerstitial (picture 4) and tubular epithelial cells (picture 3 and picture 5), and the green nuclei were less in the glomerulus (Fig. 2). We think that the larger green area in picture 4 is the rapid infiltration of inflammatory cells. In order to show the data more clearly, we have replaced picture 4 with another representative picture of this experiment group in the new manuscript.

Panel c and d: Yes, the cell vesicle in the immunohistochemical section cannot be identified under the light microscope, and only the cellular region can be distinguished. That NGAL and IL-1 β mainly express in the cytoplasm is a more suitable description. We have made the corresponding modifications in the new manuscript according to your suggestions.

Figure 5:

Panel d: The arrows in Figure 5 panel d point at lipid droplets. We have explained it in legend of Figure 5 d in the new manuscript (Line 1031).

Panel f: We are sorry to confuse you. 4-HNE positive staining is in the cytoplasm. 4-HNE is highly expressed in diabetic kidney tissue and may lead to excessive staining.

Panel g: The arrows in Figure 5 panel g point at positive DHE staining. We have explained it in legend of Figure 5 g in the new manuscript (Line 1038).

Figure 6: We are sorry for our carelessness. Podocytes are specialized epithelial cells that separate the capillary network in the glomerulus from Bowman's space. Podocytes in immunohistochemical sections are difficult to be identified by morphological features, but can be labeled and located by immunohistochemistry and immunofluorescence technique with the molecular marker WT1 (Figure 2e). It is not rigorous for us to speculate that these molecules are expressed in podocytes based on morphological features. So, we do not emphasize that these molecules are expressed in podocytes in the new manuscript. Thank you for pointing out this defect.

We are not sure whether the tubule collapse mentioned here is a tubule atrophy. However, morphologically, there seems to be no clear and definite collapse phenotype. We think that the absence of obvious staining in a few regions may be related to the difference in LDLR expression in cells. In order to eliminate this confusion, we have replaced picture 4 of Figure 6 panel c with another representative picture of this experiment group in the new manuscript.

Figure 7: This question is similar to that in Figure 6. Our explanation for podocyte confusion in the pictures of Figure 7 is as described above. The arrows in the pictures point at the positive staining of the molecules, such as SREBP1, SCD1 or FAS, which has been supplemented in Line 1064 to Line 1068 of the new manuscript.

Figure 8: SREBP-1 is a widely expressed transcription factor synthesized as a 125kd precursor that attaches to the endoplasmic reticulum and nuclear membrane. Then, SREBP-1 can be transported to the nucleus. The subcellular distribution of SREBP-1 in the cytoplasm and nucleus is closely related to a variety of proteins, which determines their intracellular translocation and

stability, and also regulates their activities as transcriptional factors (FEBS J. 2009 Feb;276(3):622-7.). We observed the different subcellular distribution of SREBP-1 by immunohistochemistry (Fig. 7a) and immunocytochemical fluorescence analysis (Fig. 8a) with the same antibody for SREBP-1. The data showed that SREBP-1 was distributed in the nucleus and cytoplasm in tissue samples (Fig7 a), but SREBP-1 was mainly distributed in the cytoplasm in cells cultured *in vitro* (fig8 a). However, the different subcellular distribution of SREBP-1 *in vivo* and *in vitro* did not affect our analysis of SREBP-1 overall expression. Influenced by the data in Fig.7a, we describe it as "SREBP-1 is mainly expressed on the cytoplasm and nucleus of HK-2 cells" in the legend in Fig. 8a. This statement is not fully accurate. Therefore, we have revised the legend of Fig. 8a as "SREBP-1 is mainly expressed in the cytoplasm of HK-2 cells".

Figure 9: panel c and d: Thank you for your important advice. We have made a brief description of Figure 9 panel c and d in the result section (Line318-324). Following your suggestion, we have detailed the differences between the groups of Figure 9 panel c and d in the new manuscript, as follows:

Compared with the WT group, the glomerular mesangial of the mice in the STZ group had a significant expansion, and renal fibrosis was significantly aggravated, and that in the STZ+ATO10 group was even more pronounced. In contrast, the glomerular mesangial expansion of STZ+ATO10 group in LDLR^{-/-} and srebp1-deficient mice was not significantly changed compared with STZ group (Fig.9c & 9d).

(2) Arrows should be added to all pictures where relevant. Otherwise, difficult to follow for the reader.

Second-round RESPONSE: Thank you for your suggestion. Arrows have added to all pictures to improve the accuracy and visual presentation of the data.

Please also describe all arrows. What do they point at?

Third-round RESPONSE: Thank you for your suggestions to improve the accessibility of our manuscript. In the Figures, we have used the arrow to point at the typical positive results, and have added the specific connotations of all arrows in the legends.

(3) You need to describe the labeling (ab) or staining – which tubules are labelled and the subcellular localizations, which areas of the ECM. This is extremely important. Please include references for the antibodies in the text.

Second-round RESPONSE: Thank you for your important suggestions. We have supplemented the labeling (ab) or staining information in the text of the new manuscript. In addition, we have summarized a table with a list of IHC antibodies, including which tubules are labelled and the subcellular localizations, references. Please review the Supplementary Table 5 for details. **For supl table 5. I would include the references and the labeling description in the text and not in a table. Please be aware that it is not possible to identify lysosomes, golgi, ER etc on light microscopy.**

Third-round RESPONSE: Thank you for your suggestions. We agree with you that it is better to illustrate the references and labeling description of all antibodies in the text than in the table. The reason why we choose to use the table way to display the information of all antibodies is because of the publication requirements of the journal, especially the restrictions on word count and number of references. Therefore, we only describe the localization of antibodies in figure legend, and no references are cited in legends. Following your important suggestions, we have supplemented the positive expression position for each antibody and described the subcellular localization of some antibodies. However, as you pointed out, lysosomes, Golgi apparatus and endoplasmic reticulum cannot be identified under light microscope. Therefore, for some antibodies, we described their expression in the cytoplasmic region or

the nuclear region in the revised manuscript, but could not clearly locate on which subcellular organelles, such as lysosomes, Golgi apparatus, endoplasmic reticulum, etc.

(4) What is HFD the abbreviation for?

Third-round RESPONSE: Thanks for your help. We feel sorry for our carelessness. HFD is the abbreviation of high-fat diet. We have supplemented the full name of HFD in the revised manuscript (Line 133).

(5) The authors write line 235 “The positive area was quantified with Image J....” Please write in details how the quantification of labeling has been performed. What are the criteria for selection of labeled tubules, criteria for background identification, are the same kind of tubules e.g., proximal tubules selected on the respective sections.

Third-round RESPONSE: Thank you for your important advice. Detailing the methods and criteria for quantitative analysis of graphs will help improve the readability of the article. Following your suggestion, we have supplemented the relevant description of graphic quantitative analysis in the Methods section of the new manuscript (Line 642 – 654), as follows:

“Integrated optical density (IOD) values of immunohistochemistry sections were evaluated by using image pro plus version 6.0 software (Media Cybernetics, Inc. , Rockville, MD, USA). The IOD of the digital image (magnification, X400) was designated as the representative RAGE staining intensity. Since RAGE is widely expressed in kidney and not specifically expressed in a particular tubule, we selected 10 randomly selected fields including distal tubules, proximal tubules and glomeruli with blind method. The IOD of each field was counted and the data were subjected to statistical analysis. Positive areas were semi-quantitatively analyzed by Image J software program (National Institutes of Health, Bethesda, MD, USA). Positive areas of the digital images (magnification, X200, and Nephryn staining X1000) were designated as representative for

calculating the value of staining intensity. The positive areas from ten randomly selected fields were calculated by blind method and subjected to statistical analysis. WT1 positive cells from 20 randomly selected fields of glomeruli sections were counted by blind method and subjected to statistical analysis.”

In all, I found the reviewer’s comments to be quite helpful, and I have revised the paper point-by-point. Again, we greatly appreciate your comments, which are very valuable in improving the quality of our manuscript.

Thank you once again for your time and consideration.

REVIEWER COMMENTS

Reviewer #2 (Remarks to the Author):

Figure 1: We agree with the reviewer's comment that distal tubules do not have a brush border. Therefore, the words "plasma membrane" in Line 941 has been replaced by "plasma membrane domains" in the revised manuscript as suggested.

The fact that distal tubules do not have a brush border has nothing to with the fact that the plasma membrane cannot be seen at LM. Fine that plasma membrane has been replaced with plasma membrane domains. In the text you now write renal tubular cells, which is fine. But on the figure is still written DT, but it look like the tubules have a brush border?

Figure 5:

Panel d: The arrows in Figure 5 panel d point at lipid droplets. We have explained it in legend of Figure 5 d in the new manuscript (Line 1037).

How can you see the actual lipid droplets in panel d? You can only see the red color corresponding to lipid?

Panel f: We are sorry to confuse you. 4-HNE positive staining is in the cytoplasm. 4-HNE is highly expressed in diabetic kidney tissue and may lead to excessive staining.

This is almost impossible to see on the pictures. A higher magnification of the pictures is required. It will also be easier to see the tubule staining if the tubules are not collapsed.

Figure 6: We are sorry for our carelessness. Podocytes are specialized epithelial cells that separate the capillary network in the glomerulus from Bowman's space. Podocytes in immunohistochemical sections are difficult to be identified by morphological features, but can be labeled and located by immunohistochemistry and immunofluorescence technique with the molecular marker WT1 (Figure 2e). It is not rigorous for us to speculate that these molecules are expressed in podocytes based on morphological features. So, we do not emphasize that these molecules are expressed in podocytes in the new manuscript. Thank you for pointing out this defect.

We are not sure whether the tubule collapse mentioned here is a tubule atrophy. However, morphologically, there seems to be no clear and definite collapse phenotype. We think that the absence of obvious staining in a few regions may be related to the difference in LDLR expression in cells. In order to eliminate this confusion, we have replaced picture 4 of Figure 6 panel c with another representative picture of this experiment group in the new manuscript.

<The collapsed tubules is normally due to the fixation. Try to illustrate the staining in tubules which are not collapsed – if possible. And please show higher magnification of the panel b-d.>

(3) You need to describe the labeling (ab) or staining – which tubules are labelled and the subcellular localizations, which areas of the ECM. This is extremely important. Please include references for the antibodies in the text.

Second-round RESPONSE: Thank you for your important suggestions. We have supplemented the labeling (ab) or staining information in the text of the new manuscript. In addition, we have summarized a table with a list of IHC antibodies, including which tubules are labelled and the subcellular localizations, references. Please review the Supplementary Table 5 for details.

For suppl table 5. I would include the references and the labeling description in the text and not in a table. Please be aware that it is not possible to identify lysosomes, golgi, ER etc on light microscopy.

Third-round RESPONSE: Thank you for your suggestions. We agree with you that it is better to illustrate the references and labeling description of all antibodies in the text than in the table. The reason why we choose to use the table way to display the information of all antibodies is because of the publication requirements of the journal, especially the restrictions on word count and number of references. Therefore, we only describe the localization of antibodies in figure legend, and no references are cited in legends. Following your important suggestions, we have supplemented the positive expression position for each antibody and described the subcellular localization of some antibodies. However, as you pointed out, lysosomes, Golgi apparatus and endoplasmic reticulum cannot be identified under light microscope. Therefore, for some antibodies, we described their expression in the cytoplasmic region or the nuclear region in the revised manuscript, but could not clearly locate on which subcellular organelles, such as lysosomes, Golgi apparatus, endoplasmic reticulum, etc.

<Please include references in the text, where the labeling is described.>

(5) The authors write line 235 “The positive area was quantified with Image J...” Please write in details how the quantification of labeling has been performed. What are the criteria for selection of labeled tubules, criteria for background identification, are the same kind of tubules e.g., proximal tubules selected on the respective sections.

Third-round RESPONSE: Thank you for your important advice. Detailing the methods and criteria for quantitative analysis of graphs will help improve the readability of the article. Following your suggestion, we have supplemented the relevant description of graphic quantitative analysis in the Methods section of the new manuscript (Line 646 – 661), as follows:

“Integrated optical density (IOD) values of immunohistochemistry sections were evaluated by using image pro plus version 6.0 software (Media Cybernetics, Inc. , Rockville, MD, USA). The IOD of the digital image (magnification, X400) was designated as the representative RAGE staining intensity. Since RAGE is widely expressed in kidney and not specifically expressed in a particular tubule, we selected 10 randomly selected fields including distal tubules, proximal tubules and glomeruli with blind method. The IOD of each field was counted and the data were subjected to statistical analysis. Positive areas were semi- quantitatively analyzed by Image J software program (National Institutes of Health, Bethesda, MD, USA). Positive areas of the digital images (magnification, X200, and Nephryn staining X1000) were designated as representative for calculating the value of staining intensity. The positive areas from ten randomly selected fields were calculated by blind method and subjected to statistical analysis. WT1 positive cells from 20 randomly selected fields of glomeruli sections were counted by blind method and subjected to statistical analysis.”

How does this methods take potential background staining into account?

We truly appreciate the editors and reviewers for their time and effort in reviewing our manuscript. Your comments provided valuable suggestions for further revision and improvement of our manuscript. In this submission, some revisions were made based on the comments from editor and reviewers. Descriptions of these changes are underlined in the manuscript. Response to the comments is addressed below:

Reviewer #2:

(1) Figure 1: Question: The fact that distal tubules do not have a brush border has nothing to with the fact that the plasma membrane cannot be seen at LM. Fine that plasma membrane has been replaced with plasma membrane domains. In the text you now write renal tubular cells, which is fine. But on the figure is still written DT, but it looks like the tubules have a brush border?

Third-round RESPONSE: Thank you for your careful and rigorous review. I'm sorry for my carelessness. In Figure 1g, it should be PT, not DT. We have corrected it in Figure 1 of the new document.

(2) Figure 5:

Panel d: Question: How can you see the actual lipid droplets in panel d? You can only see the red color corresponding to lipid?

Third-round RESPONSE: Oil Red O is a fat-soluble dye that stains neutral triglycerides and lipids. We agree with your suggestion that the positive staining of oil red O should be described as lipid, not lipid droplets. We have corrected it in legend of Figure 5 d in the new manuscript (Line 1073).

Panel f: Question: This is almost impossible to see on the pictures. A higher magnification of the pictures is required. It will also be easier to see the tubule staining if the tubules are not collapsed.

Third-round RESPONSE: Thank you for pointing out the insufficiency of the pictures. In addition to the pathological changes in diabetic nephropathy itself, the fixation method can also lead to collapsed tubules. Indeed, the collapsed tubules may affect tubule staining. In order to observe the staining of tubules more easily, we have tried our best to replace the IHC pictures of 4-HNE and show the 1000x pictures with the tubules that have not collapsed in the new manuscript (Figure 5f).

(3) Figure 6: Question: The collapsed tubules is normally due to the fixation. Try to illustrate the staining in tubules which are not collapsed – if possible. And please show higher magnification of the panel b-d.

Third-round RESPONSE: There are many factors that may affect the collapse of tubules during tissue fixation, such as embedding, dehydration, fixation fluid selection, fixation time, and pressure during perfusion fixation. To reduce your confusion about the data, we have reselected the field of view to take pictures. We replaced the original Figure6 b-d with new pictures and multiplied it by 1000 times (Figure 6b-d).

(4) Question: Please include references in the text, where the labeling is described.

Third-round RESPONSE: In the last versions of our manuscript, we chose to display the references for all antibodies in tabular form because of the publication requirements of the journal, especially the limit for the number of words and references. After communicating with the journal editor, we are allowed to cite more references in the text. Therefore, we have included the references for all antibodies in the new manuscript.

(5) Question: How does these methods take potential background staining into account?

Third-round RESPONSE: We are not sure that the potential background staining you mentioned above refers to background staining or non-specific staining in immunohistochemistry (IHC) or immunofluorescence labeling (IF), or the background color of a picture. Indeed, background staining or non-specific staining is an often-encountered disadvantage in IHC and IF due to cross reaction of antibodies. In our studies, the IHC and IF staining is highly specific and the background staining is usually low. Even if there is a small amount of background staining, the effect of background staining or non-specific staining on the research conclusion can be eliminated by setting up the control groups. If it is the background color of a picture, we can eliminate the influence of the background color on the positive expression analysis by setting the ' threshold ' of Image J or image pro plus version 6.0 which are efficient software. Therefore, our quantitative statistics of IHC images are scientific and objective.

In all, we found the reviewer's comments to be quite helpful, and we have revised the paper point-by-point. Again, we greatly appreciate your comments, which are very valuable in improving the quality of our manuscript.

Thank you once again for your time and consideration.

REVIEWERS' COMMENTS

Reviewer #2 (Remarks to the Author):

(1) Figure 1: Question: The fact that distal tubules do not have a brush border has nothing to with the fact that the plasma membrane cannot be seen at LM. Fine that plasma membrane has been replaced with plasma membrane domains. In the text you now write renal tubular cells, which is fine. But on the figure is still written DT, but it looks like the tubules have a brush border?

Third-round RESPONSE: Thank you for your careful and rigorous review. I'm sorry for my carelessness. In Figure 1g, it should be PT, not DT. We have corrected it in Figure 1 of the new document.

Ok

(2) Figure 5:

Panel d: Question: How can you see the actual lipid droplets in panel d? You can only see the red color corresponding to lipid?

Third-round RESPONSE: Oil Red O is a fat-soluble dye that stains neutral triglycerides and lipids. We agree with your suggestion that the positive staining of oil red O should be described as lipid, not lipid droplets. We have corrected it in legend of Figure 5 d in the new manuscript (Line 1073).

Ok

Panel f: Question: This is almost impossible to see on the pictures. A higher magnification of the pictures is required. It will also be easier to see the tubule staining if the tubules are not collapsed.

Third-round RESPONSE: Thank you for pointing out the insufficiency of the pictures. In addition to the pathological changes in diabetic nephropathy itself, the fixation method can also lead to collapsed tubules. Indeed, the collapsed tubules may affect tubule staining. In order to observe the staining of tubules more easily, we have tried our best to replace the IHC pictures of 4-HNE and show the 1000x pictures with the tubules that have not collapsed in the new manuscript (Figure 5f).

Ok

(3) Figure 6: Question: The collapsed tubules is normally due to the fixation. Try to illustrate the staining in tubules which are not collapsed – if possible. And please show higher magnification of the panel b-d.

Third-round RESPONSE: There are many factors that may affect the collapse of tubules during tissue fixation, such as embedding, dehydration, fixation fluid selection, fixation time, and pressure during perfusion fixation. To reduce your confusion about the data, we have reselected the field of view to take pictures. We replaced the original Figure6 b-d with new pictures and multiplied it by 1000 times (Figure 6b-d).

HMGLR and LDLR are localized in proximal tubules according to the pictures – right?

But in which cells/structures are CD38 localized? Please describe this in the text.

(4) Question: Please include references in the text, where the labeling is described.

Third-round RESPONSE: In the last versions of our manuscript, we chose to display the references for all antibodies in tabular form because of the publication requirements of the journal, especially the limit for the number of words and references. After communicating with the journal editor, we are allowed to cite more references in the text. Therefore, we have included the references for all antibodies in the new manuscript.

Ok

(5) Question: How does these methods take potential background staining into account?

Third-round RESPONSE: We are not sure that the potential background staining you mentioned above refers to background staining or non-specific staining in immunohistochemistry (IHC) or immunofluorescence labeling (IF), or the background color of a picture. Indeed, background staining or non-specific staining is an often-encountered disadvantage in IHC and IF due to cross reaction of antibodies. In our studies, the IHC and IF staining is highly specific and the background staining is usually low. Even if there is a small amount of background staining, the effect of background staining or non-specific staining on the research conclusion can be eliminated by setting up the control groups. If it is the background color of a picture, we can eliminate the influence of the background color on the positive expression analysis by setting the ' threshold ' of Image J or image pro plus version 6.0 which are efficient software. Therefore, our quantitative statistics of IHC images are scientific and objective.

Please include your considerations on the potential background in the text. This is important information for the reader since you quantify larger areas of the sections and not specific tubules.

We truly appreciate the editors and reviewers for their time and effort in reviewing our manuscript. Your comments provided valuable suggestions for further revision and improvement of our manuscript. In this submission, some revisions were made based on the comments from editor and reviewers. Descriptions of these changes are underlined in the manuscript. Response to the comments is addressed below:

Reviewer #2:

(1) Question: Figure 6:

HMGCR and LDLR are localized in proximal tubules according to the pictures – right?

But in which cells/structures are CD38 localized? Please describe this in the text.

Fourth-round RESPONSE: Thank you for your careful observation and consideration. The increase of cholesterol uptake and synthesis will lead to lipid deposition in diabetic kidney, thus causing renal dysfunction. Low density lipoprotein receptor (LDLR) and 3-hydroxy-3-methylglutaryl coenzyme A reductase (HMGCR) play important roles in maintaining cholesterol uptake and synthesis¹. According to our immunohistochemical staining results and other researchers' data, HMGCR and LDLR are predominantly localized in renal tubule, especially proximal tubules, but can also be expressed in glomerular mesangial cells²⁻⁴.

I'm not sure whether the CD38 you mentioned above is the CD36 in our text. CD36 is a multifunctional receptor for long-chain fatty acids, oxidized lipids, advanced oxidation protein products, thrombospondin and advanced glycation end products. CD36 is expressed in a wide variety of kidney cells such as proximal tubular epithelial cells, mesangial cells, podocytes, monocytes and macrophages⁵, and mainly expressed on the cell membrane. Therefore, besides proximal tubules, CD36 can also be highly expressed in glomeruli, which is different from HMGCR and LDLR. We have described this in the text (Line 1139-1142).

(2) Question: Please include your considerations on the potential background in the text. This is important information for the reader since you quantify larger areas of the sections and not specific tubules.

Fourth-round RESPONSE: Thank you for your reminding. This is indeed important information for the reader to understand our quantitative methodology, and we have added it in the text (Line 715-720).

In all, we found the reviewer's comments to be quite helpful, and we have revised the paper point-by-point. Again, we greatly appreciate your comments, which are very valuable in improving the quality of our manuscript.

Thank you once again for your time and consideration.

Reference

- 1 Sun, H., Yuan, Y. & Sun, Z. L. Cholesterol Contributes to Diabetic Nephropathy through SCAP-SREBP-2 Pathway. *Int J Endocrinol* **2013**, 592576, doi:10.1155/2013/592576 (2013).
- 2 Xu, Z. E. *et al.* Inflammatory stress exacerbates lipid-mediated renal injury in ApoE/CD36/SRA triple knockout mice. *Am J Physiol Renal Physiol* **301**, F713-722, doi:10.1152/ajprenal.00341.2010 (2011).
- 3 Zheng, Y. *et al.* Anti-Inflammatory Effects of Ang-(1-7) in Ameliorating HFD-Induced Renal Injury through LDLr-SREBP2-SCAP Pathway. *PLoS One* **10**, e0136187, doi:10.1371/journal.pone.0136187 (2015).
- 4 Chen, X. *et al.* Disulfide-bond A oxidoreductase-like protein protects against ectopic fat deposition and lipid-related kidney damage in diabetic nephropathy. *Kidney Int* **95**, 880-895, doi:10.1016/j.kint.2018.10.038 (2019).
- 5 Yang, X. *et al.* CD36 in chronic kidney disease: novel insights and therapeutic opportunities. *Nat Rev Nephrol* **13**, 769-781, doi:10.1038/nrneph.2017.126 (2017).